# Error-quantified Conformal Inference for Time Series

**Junxi Wu**[1,2]*, **Dongjian Hu**[1]*, **Yajie Bao**[1]†, **Shu-Tao Xia**[2]†, **Changliang Zou**[1]

[1] School of Statistics and Data Science, LPMC, KLMDASR and LEBPS, Nankai University
[2] Tsinghua Shenzhen International Graduate School, Tsinghua University
{wujunxi,hudongjian}@mail.nankai.edu.cn
{yajiebao,zoucl}@nankai.edu.cn, xiast@sz.tsinghua.edu.cn

## Abstract

Uncertainty quantification in time series prediction is challenging due to the temporal dependence and distribution shift on sequential data. Conformal inference provides a pivotal and flexible instrument for assessing the uncertainty of machine learning models through prediction sets. Recently, a series of online conformal inference methods updated thresholds of prediction sets by performing online gradient descent on a sequence of quantile loss functions. A drawback of such methods is that they only use the information of revealed non-conformity scores via miscoverage indicators but ignore error quantification, namely the distance between the non-conformity score and the current threshold. To accurately leverage the dynamic of miscoverage error, we propose *Error-quantified Conformal Inference* (ECI) by smoothing the quantile loss function. ECI introduces a continuous and adaptive feedback scale with the miscoverage error, rather than simple binary feedback in existing methods. We establish a long-term coverage guarantee for ECI under arbitrary dependence and distribution shift. The extensive experimental results show that ECI can achieve valid miscoverage control and output tighter prediction sets than other baselines.

## 1 Introduction

Uncertainty quantification for time series is crucial across various domains including finance, climate science, epidemiology, energy, supply chains, and macroeconomics, etc, especially in high-stakes areas. To achieve this goal, an ideal model is supposed to consistently produce prediction sets that are well-calibrated, meaning that over time, the proportion of sets containing true labels should align closely with the intended confidence level. Classic methods for uncertainty quantification often rely on strict parametric assumptions of time-series models like autoregressive and moving average (ARMA) models (Brockwell & Davis, 1991). Other methods like Bayesian recurrent neural networks (Fortunato et al., 2017) and deep Gaussian processes (Li et al., 2020) are difficult to calibrate by themselves. And quantile regression models (Gasthaus et al., 2019) may "overfit" when estimating uncertainty. Additionally, complex machine learning models such as transformer (Vaswani et al., 2017; Li et al., 2019; Gao et al., 2024) have been designed to output accurate predictions but cannot provide valid prediction sets. Hence, a systematic tool is required to perform uncertainty quantification for complex black-box models in time series data.

Conformal inference (Vovk et al., 2005) is an increasingly popular framework for uncertainty quantification with arbitrary underlying point predictors (whether statistical, machine or deep learning). At its core, conformal prediction sets are guaranteed to contain the true label with a specified probability, under solely the assumption that the data is exchangeable. This is achieved without making parametric assumptions about the underlying data distribution, thereby enhancing its applicability across a wide range of models and datasets. However, in time series data, exchangeability does not hold due to strong correlations and potential distribution shifts.

---

*Equal contribution.
†Corresponding authors

Recently, a growing body of research has focused on developing online conformal methods for scenarios where data arrives sequentially. One of the most important branches is the Adaptive Conformal Inference (ACI) proposed by Gibbs & Candès (2021), and its following works (Zaffran et al., 2022; Bhatnagar et al., 2023; Gibbs & Candès, 2024). At each time step, these methods generate a prediction set characterized by a single threshold or confidence parameter that regulates the size of the set, for example $\hat{C}_t = \{y \in \mathcal{Y} : S_t(X_t, y) \leq q_t\}$, where $S_t(\cdot, \cdot)$ is the non-conformity score function. After $Y_t$ is observed, they update the threshold $q_t$ through indicator $\mathbb{1}\{Y_t \notin \hat{C}_t\}$, which is identical to $\mathbb{1}\{S_t(X_t, Y_t) > q_t\}$. However, simple binary feedback cannot precisely capture the magnitude of error[1] $S_t(X_t, Y_t) - q_t$, quantifying the extent of under/over coverage of $\hat{C}_t$. For example, in the miscoverage case, an empty prediction set and a prediction set that almost covers the true label yield the same feedback value in ACI and its variants. Hence it will take a longer time to correct past mistakes, see the blue curve in Figure 1.

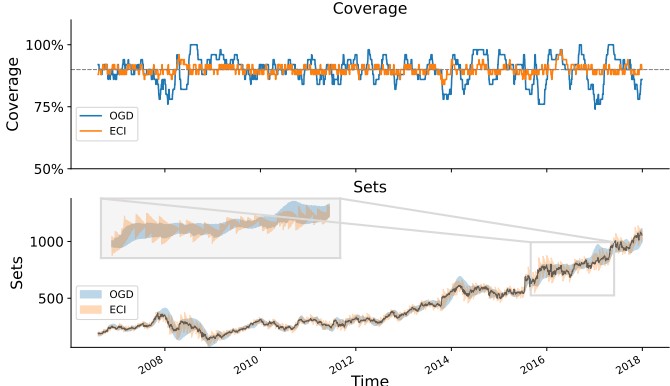

Figure 1: Comparison results between OGD (online (sub)gradient descent) and ECI on Google stock dataset with Prophet model. OGD uses the same feedback as ACI but updates the confidence level. The coverage is averaged over a rolling window of 50 points.

In this paper, we propose *Error-quantified Conformal Inference* (ECI) based on an adaptive *error quantification* (EQ) term that provides additional smooth feedback. ECI not only uses the miscoverage indicator in the feedback, but also introduces a continuous EQ function to assess the magnitude of revealed error $S_t(X_t, Y_t) - q_t$. Benefiting from the EQ term, our online conformal procedure will react quickly to distribution shifts in time series, see Figure 1. This leads to tighter prediction sets without sacrificing the miscoverage rate. We summarize our main contributions as follows:

- We propose ECI for uncertainty quantification in time series. It is a novel method based on adaptive updates with additional smooth feedback, quantifying the extent of under/over coverage. We further propose two variants of ECI. The first introduces a cutoff threshold for the EQ term to avoid over-compensation caused by small errors. Another variant integrates the error of previous steps to make coverage more stable.

- There are mainly two theoretical results. Firstly, we obtain a coverage guarantee for ECI with a fixed learning rate and not restricted to long-term. We prove it by showing that every miscoverage step will be followed by several coverage steps given a proper learning rate. Secondly, for arbitrary learning rates, we give a finite-sample upper bound for the averaged miscoverage error. Both theoretical results do not need any assumption on the data-generating distribution.

- Extensive experimental results demonstrate that ECI and its variants provide superior performance in time series, including data in finance, energy, and climate domains. We show that ECI maintains coverage at the target level and obtains tighter prediction sets than other state-of-the-art methods.

---

[1]To distinguish from the miscoverage error $\mathbb{1}\{S_t(X_t, Y_t) > q_t\} - \alpha$, we refer "error" to the term $S_t(X_t, Y_t) - q_t$ in the rest of our paper.

## 2 BACKGROUND AND PROBLEM SETUP

### 2.1 CONFORMAL INFERENCE

Let $\hat{f} : \mathcal{X} \to \mathcal{Y}$ be a prediction model trained on an independent training set. Given labeled data $\{(X_i, Y_i)\}_{i \leq n} \subset \mathcal{X} \times \mathcal{Y}$ and test data $X_{n+1} \in \mathcal{X}$, the objective is to construct a confidence set for the unknown label $Y_{n+1}$. To determine whether the candidate value $y$ is a reasonable estimate of $Y_{n+1}$, we define a non-conformity score function $S(\cdot, \cdot) : \mathcal{X} \times \mathcal{Y} \to \mathbb{R}$. For example, in the regression task, we may take absolute residual score, scaled residual score (Lei et al., 2018), and conformalized quantile regression score (Romano et al., 2019). Generally, $S(X_i, Y_i)$ depends on the base model $\hat{f}$ and measures how well the prediction value $\hat{f}(X_i)$ conforms the true label $Y_i$. Given the nominal level $\alpha \in (0, 1)$, split conformal prediction (Papadopoulos et al., 2002; Lei et al., 2018) outputs the prediction set $\hat{C}(X_{n+1}) = \{y \in \mathcal{Y} : S(X_{n+1}, y) \leq s\}$, where the threshold $s$ is the $\lceil (1 - \alpha)(n + 1) \rceil$-th smallest value among $\{S(X_i, Y_i)\}_{i=1}^n$. If $\{(X_i, Y_i)\}_{i=1}^{n+1}$ are exchangeable, we have the finite-sample coverage guarantee $\mathbb{P}\{Y_{n+1} \in \hat{C}(X_{n+1})\} \geq 1 - \alpha$.

In practice, split conformal prediction divides the labeled data into a training set for fitting the predictive model and a calibration set for computing non-conformity scores. There are other variants of conformal inference to better utilize the labeled data, like full conformal (Vovk et al., 2005), Jackknife+ (Barber et al., 2021). In many scenarios, data may exhibit a distribution shift and thus are no longer exchangeable. Chernozhukov et al. (2018) extend conformal inference to ergodic cases with dependent data, but needs transformations of data to be a strong mixing series. Oliveira et al. (2024) study split conformal prediction for non-exchangeable data relying on some distributional assumptions. We refer to Tibshirani et al. (2019), Podkopaev & Ramdas (2021), Barber et al. (2023) and Yang et al. (2024a) for more development dealing with non-exchangeable data.

### 2.2 CONFORMAL INFERENCE FOR SEQUENTIAL DATA

Recently, significant efforts have been made to extend conformal inference to online schemes. Gibbs & Candès (2021) proposed ACI which models the distribution shift in time series as a learning problem in a single parameter whose optimal value varies over time. Based on ACI, several works (Gibbs & Candès, 2024; Zaffran et al., 2022; Bhatnagar et al., 2023; Podkopaev et al., 2024; Angelopoulos et al., 2024; Yang et al., 2024b) used online learning techniques to adaptively adjust the size of the prediction set based on recent observations. Gibbs & Candès (2024) and Bhatnagar et al. (2023) utilized meta-learning approaches to aggregate the results updated with multiple learning rates or experts, where main algorithms were adapted from Gradu et al. (2023) and Jun et al. (2017) respectively. In addition, Podkopaev et al. (2024) extended the betting technique in Orabona & Pál (2016) to the conformal setting and proposed a new parameter-free algorithm. The methods mentioned above can achieve a long-term coverage guarantee without any assumptions about the data-generating process. Xu & Xie (2021; 2023) casted the problem of constructing a conformal prediction set as predicting the quantile of a future residual and propose algorithms to adaptively re-estimate (conditional) quantiles. However, these methods are constrained by model and distribution assumptions and potentially suffer from over-fitting problems. Weinstein & Ramdas (2020) and Bao et al. (2024) investigated online selective conformal inference problem for i.i.d. data stream.

Closely related to our work is the Conformal PID algorithm of Angelopoulos et al. (2023b), which simplifies and strengthens existing analyses in online conformal inference with ideas from control theory. Our algorithm differs from theirs by replacing the integration and scorecasting with the EQ term, and is thus able to yield significantly tighter prediction sets with valid coverage. A more detailed description of the competing methods is included in the Appendix C.

### 2.3 PROBLEM SETUP

Suppose the time series data $\{(X_t, Y_t)\}_{t \geq 1} \subset \mathcal{X} \times \mathcal{Y}$ are collected sequentially. At time $t$, our goal is to construct a prediction set $\hat{C}_t(X_t)$ for the unseen label $Y_t$ based on the machine learning model trained on previously observed data $\{(X_i, Y_i)\}_{i \leq t}$. Aligned with the standard conformal inference methods, we use a non-conformity score function $S_t(\cdot, \cdot) : \mathcal{X} \times \mathcal{Y} \to \mathbb{R}$ that may change over time. For example, in the regression task, $S_t(x, y) = |y - \hat{f}_t(x)|$ with the base model $\hat{f}_t$ trained at time $t$. Then we construct the conformal prediction set by

$$\hat{C}_t(X_t) = \{y \in \mathcal{Y} : S_t(X_t, y) \leq q_t\}, \tag{1}$$

where $q_t$ is the threshold that estimates or approximates $(1 - \alpha)$-th quantile for the distribution of the non-conformity score $S_t(X_t, Y_t)$.

Note that if the data sequence $\{(X_i, Y_i)\}_{i \leq t+1}$ are i.i.d. or exchangeable, according to the split conformal prediction, taking $q_t$ in Equation (1) to be the $(1 - \alpha)(1 + t^{-1})$-th sample quantile of $\{S_i(X_i, Y_i)\}_{i \leq t}$ yields a valid coverage guarantee $\mathbb{P}\{Y_{t+1} \in \hat{C}_{t+1}(X_{t+1})\} \geq 1 - \alpha$. However, in an adversarial setting where exchangeability does not hold, such as time series data with strong correlations or distribution shifts, it is very difficult to achieve such a real-time coverage guarantee. Recent works (Bhatnagar et al., 2023; Angelopoulos et al., 2023a; 2024; Podkopaev et al., 2024) began to consider the following long-term miscoverage control:

$$\lim_{T \to \infty} \frac{1}{T} \sum_{t=1}^{T} \mathbb{1}\{Y_t \notin \hat{C}_t(X_t)\} = \alpha. \tag{2}$$

Hence, our primary target is to design an online algorithm to dynamically choose thresholds $\{q_t\}_{t \geq 1}$ that can achieve the control in (2) without any distributional assumptions about data.

## 3 ERROR-QUANTIFIED CONFORMAL INFERENCE

### 3.1 ERROR QUANTIFICATION VIA SMOOTHING FEEDBACK

The online conformal inference framework developed by Gibbs & Candès (2021) adopts the online learning technique to learn the thresholds $\{q_t\}_{t \geq 1}$ based on previously observed data. Let $s_t = S(X_t, Y_t)$ and $\ell_t(q) = (s_t - q)(\mathbb{1}\{s_t > q\} - \alpha)$ denotes the $(1 - \alpha)$-th quantile loss. When the label $Y_t$ is revealed, we can perform online (sub)gradient descent (OGD) as

$$q_{t+1} = q_t - \eta \, \nabla \ell_t(q_t) = q_t + \eta(\text{err}_t - \alpha), \tag{3}$$

where $\text{err}_t = \mathbb{1}\{s_t > q_t\} = \mathbb{1}\{Y_t \notin \hat{C}_t(X_t)\}$ is the miscoverage indicator and $\eta$ is the learning rate. The subgradient $\text{err}_t - \alpha$ can be regarded as the feedback after observing label $Y_t$: if $\hat{C}_t(X_t)$ does not cover $Y_t$ (i.e., $\text{err}_t = 1$), OGD will increase the threshold to construct a more liberal prediction set in the next step; otherwise, OGD will construct a more conservative prediction set. In fact, ACI uses the same idea but updates the confidence level of the prediction set instead. Notice that the enrolled average of history feedback $\sum_{t=1}^{T}(\text{err}_t - \alpha)/T$ is exactly the averaged miscoverage we aim to control in Equation (2). Essentially, the long-term coverage guarantee of OGD or ACI comes from the equivalence between feedback and control.

However, subgradients of quantile loss $\ell_t$ can only take two values $1 - \alpha$ and $-\alpha$, regarding the feedback value of coverage ($s_t \leq q_t$) and miscoverage ($s_t > q_t$) respectively. As a consequence, the feedback keeps the same value no matter how severe the miscoverage is or how conservative the coverage is. In other words, discrete feedback value does not exploit the information of the error.

To address this issue, we consider smoothing the feedback when updating the thresholds. Let $f(x) \in [0, 1]$ be a smooth approximation to the indicator $\mathbb{1}(x > 0)$, such as Sigmoid function, Gaussian error function, etc. Then we can approximate the quantile loss $\ell_t(q)$ via $\tilde{\ell}_t(q) = (f(s_t - q) - \alpha)(s_t - q)$. This smoothing technique was demonstrated in previous studies (Kaplan & Sun, 2017; Fernandes et al., 2021). Applying OGD on the smoothed quantile loss $\{\tilde{\ell}_t(q)\}_{t \geq 1}$, we have the following fully smoothed update rule

$$q_{t+1} = q_t - \eta \nabla \tilde{\ell}_t(q_t) = q_t + \eta \cdot \left[ f(s_t - q_t) - \alpha + (s_t - q_t)\nabla f(s_t - q_t) \right]. \tag{4}$$

Here we called $(s_t - q_t)\nabla f(s_t - q_t)$ as the *error quantification* (EQ) term, which provides additional feedback based on the magnitude of error $s_t - q_t$.

To have a preliminary view of the effect of EQ term, we take Sigmoid function $f(x) = \frac{1}{1+\exp(-cx)}$ as an example, and show the curve of EQ function $x\nabla f(x)$ in Figure 2. Firstly, the feedback from the EQ term has the same sign as subgradient $\text{err}_t - \alpha$ by letting $x = s_t - q_t$, which means that it does not deviate from the direction of update in OGD. On the other hand, when $s_t$ significantly deviates from the current threshold $q_t$ (for example, an abrupt change point appears at time $t$), the EQ term tends to decrease as $s_t - q_t$ grows. This is to prevent the subsequent prediction sets from running out of control due to a single anomaly data point.

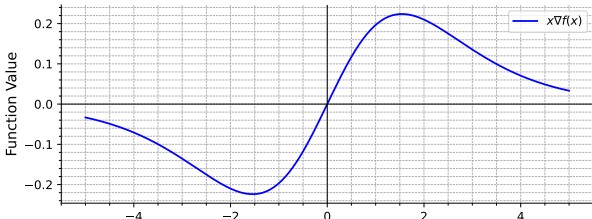

Figure 2: Dynamics of the EQ function across variable $x$, where $f(x)$ is Sigmoid function and $c = 1$.

However, Equation (4) may introduce extra bias during the derivation process, as detailed in Appendix A.1. If we perform the fully smoothed update rule (4), we cannot directly control the averaged miscoverage gap $\sum_{t=1}^{T} \text{err}_t / T - \alpha$ due to the smoothing bias.

Hence we keep the EQ term and replace $f(s_t - q_t)$ in Equation (4) with the miscoverage indicator $\text{err}_t$, that is

$$q_{t+1} = q_t + \eta \cdot \left[\text{err}_t - \alpha + (s_t - q_t)\nabla f(s_t - q_t)\right]. \tag{5}$$

Formally, we propose *Error-quantified Conformal Inference* (ECI) based on (5), which leverages the miscoverage indicator and the EQ term simultaneously to update thresholds with partially smoothed feedback. Compared with OGD in Equation (3), the feedback from EQ term is compensation over the original subgradient $\text{err}_t - \alpha$. Clearly, for the same pair $(s_t, q_t)$, ECI can provide more feedback compared with OGD and fully smoothed rule in Equation (4). We provide an ablation study in Appendix A.2 to illustrate the importance of the EQ term.

## 3.2 EXTENDED VERSIONS

**ECI-cutoff.** The EQ term reacts quickly to sudden distribution shifts like changepoints. However, when $q_t$ is slightly larger/smaller than $s_t$, the additional adjustment provided by the EQ term may lead to under/over coverage of prediction sets. For example, if $s_t < q_t$ and $s_{t+1} \in [q_t - \eta_t \alpha + \eta_t(s_t - q_t)\nabla f(s_t - q_t), q_t - \eta_t \alpha]$, then adding the EQ term will cause miscoverage, i.e. $s_{t+1} > q_{t+1}$. Therefore, we introduce a cutoff to our added term:

$$q_{t+1} = q_t + \eta \cdot \left[(\text{err}_t - \alpha) + (s_t - q_t)\nabla f(s_t - q_t)\mathbb{1}(|s_t - q_t| > h_t)\right], \tag{6}$$

where $h_t = h \max_{t-w \leq i,j \leq t} |s_i - s_j|$, and $h$ is a pre-determined threshold, $w$ is window length.

**ECI-integral.** More information input generally means better performance. Note that (5) only takes the information of one last timestep into consideration. A natural alternative is to integrate the error of more than a single step. Therefore, we propose the extended update:

$$q_{t+1} = q_t + \eta \cdot \sum_{i=1}^{t} w_i \left\{\text{err}_i - \alpha + (s_i - q_i)\nabla f(s_i - q_i)\right\}, \tag{7}$$

where $\{w_i\}_{1 \leq i \leq t} \subseteq [0, 1]$ is a sequence of increasing weights with $\sum_{i=1}^{t} w_i = 1$. Equation (7) evaluates the recent empirical miscoverage frequency and degree of miscovery when deciding whether or not to lower or raise $q_t$, making coverage more stable.

## 3.3 DISTRIBUTION-FREE COVERAGE GUARANTEES

In this section, we outline the theoretical coverage guarantees of ECI. The detailed proofs are referred in Appendix B. We first present two assumptions and briefly explain their feasibility.

**Assumption 1.** *For any $t \in \mathbb{N}_+$, there exists $B > 0$ such that $s_t \in [0, B]$.*

**Assumption 2.** $|x\nabla f(x)| \leq \lambda$ and $|\nabla f(x)| \leq c$ *for any $x \in \mathbb{R}$, where $\lambda, c > 0$ are constants.*

Assumption 1 assumes boundedness of scores, which is ubiquitous in online conformal literature (Gibbs & Candès, 2021; Angelopoulos et al., 2023b; 2024). For Assumption 2, note that a typical choice of $f$ is the Sigmoid function $\sigma(cx)$ with a scale parameter $c > 0$. Then

$$|\nabla f(x)| = c \cdot |\sigma(cx)(1 - \sigma(cx))| \leq \frac{c}{4},$$

$$|x\nabla f(x)| = |cx\sigma(cx)(1 - \sigma(cx))| < |cx\frac{e^{cx}}{1 + e^{cx}}| < |cxe^{cx}| \le \frac{1}{e}.$$

Thus Assumption 2 is naturally satisfied. Moreover, if we consider ECI-cutoff and replace $\nabla f(x)$ by $\nabla f_h(x) = \nabla f(x)\mathbb{1}(|x| > h)$, then

$$|\nabla f_h(x)| = c \cdot |\sigma(cx)(1 - \sigma(cx))|\mathbb{1}(|x| > h) < \frac{c}{1 + e^{ch}},$$

which can be sufficiently small as long as we set $c$ to be large.

The following theorem provides a dynamic miscoverage bound when the learning rate is fixed. We prove it by showing that every miscoverage step will be followed by at least $N - 1$ coverage step, where $N = \lfloor \alpha^{-1} \rfloor$, that is if $Y_t \notin \hat{C}_t$ happens, then $Y_{t+i} \in \hat{C}_{t+i}$ holds for $i = 1, 2, \cdots, N - 1$.

**Theorem 1.** *Assume that $\eta > 2NB$, $c < \frac{\min\{\eta, N^2\}}{2N^2[B+(1-\alpha+\lambda)\eta]}$, where $N = \lfloor \frac{1}{\alpha} \rfloor$. Under Assumptions 1 and 2, the prediction sets generated by* (5) *satisfies for any $T$,*

$$\frac{1}{N} \sum_{t=T+1}^{T+N} \mathbb{1}\{Y_t \notin \hat{C}_t\} \le \frac{1}{N}. \tag{8}$$

As standard choices for $\alpha$ are $\{0.01, 0.05, 0.1, 0.2\}$, if we further assume $\alpha = N^{-1}$ for some $N \in \mathbb{N}$, it follows from (8) that $\frac{1}{N} \sum_{t=T+1}^{T+N} \mathbb{1}\{Y_t \notin \hat{C}_t\} \le \alpha$ for any $T$, which immediately yields the long-term coverage guarantee in Equation (2).

We present the bound of averaged miscoverage error under adaptive learning rates. The result is also distribution-free and only needs a proper choice of learning rate sequence.

**Theorem 2.** *Let $\{\eta_t\}_{t\ge1}$ be an arbitrary positive sequence. Under Assumptions 1 and 2, the prediction sets generated by* (5) *with adaptive learning rate $\eta_t$ satisfies:*

$$\left|\frac{1}{T} \sum_{t=1}^{T}(err_t - \alpha)\right| \le \frac{(B + M_{T-1})\|\Delta_{1:T}\|_1}{T} + c\left[B + (1 - \alpha + \lambda)M_{T-1}\right], \tag{9}$$

*where $\|\Delta_{1:T}\|_1 = |\eta_1^{-1}| + \sum_{t=2}^{T} |\eta_t^{-1} - \eta_{t-1}^{-1}|$, $M_T = \max_{1 \le r \le T} \eta_r$.*

For the first term, following the analysis in Angelopoulos et al. (2024), if the learning rate decreases over extended periods when the distribution appears stable, but then increases again, in a repetitive manner, $\|\Delta_{1:T}\|_1/T$ can be sufficiently small. To be specific, $\|\Delta_{1:T}\|_1 \le 2N_t/(\min_{t \le T} \eta_t)$, where $N_T = \sum_{t=1}^{T} \mathbb{1}\{\eta_{t+1} > \eta_t\}$ denotes the number of times the learning rate is increased. Hence, if $\eta_t$ does not decay too quickly and the number of "resets" $N_T$ is $o(T)$, the first term in the upper bound in (2) will be within an acceptable range. For the second term, if we set $c$ as a sufficiently small value, then the second term is also limited to a small value.

## 4 EXPERIMENTS

### 4.1 SETUP

**Datasets.** We evaluate four real-world datasets: Amazon stock, Google stock (Nguyen, 2018), electricity demand (Harries et al., 1999) and temperature in Delhi (Vrao., 2017). Besides, we evaluate the synthetic dataset under changepoint setting. In the subsequent sections, we will provide a detailed introduction to each of these datasets.

**Base predictors.** We consider three diverse types of base predictors.

- Prophet (Taylor & Letham, 2018): As a Bayesian additive model, Prophet predicts the value $\hat{Y}_t$ as a function of time $t$, expressed as $\hat{Y}_t = g(t) + s(t) + h(t) + \epsilon_t$, where $g(t)$ models the overall trend, $s(t)$ accounts for periodic seasonal effects, $h(t)$ captures holiday effects, and $\epsilon_t$ represents the noise assumed to follow a normal distribution.

- AR (AutoRegressive Model): As a classic model, AR is defined as $\hat{Y}_t = \phi_1 Y_{t-1} + \phi_2 Y_{t-2} + \ldots + \phi_p Y_{t-p} + \epsilon_t$, where $\phi_1, \phi_2, ..., \phi_p$ are the parameters, and $p$ is the order of AR model. Following Angelopoulos et al. (2023b), we set $p = 3$.
- Theta (Assimakopoulos & Nikolopoulos, 2000): As a decomposition-based forecasting approach, Theta model modify the curvature of a time series by applying a coefficient $\theta$ to its second differences. We typically use $\theta = 0$ (the long-term trend) and $\theta = 2$ (the short-term dynamics) to decompose the original series into two components.

**Baselines.** We compare with four state-of-the-art methods: ACI (Gibbs & Candès, 2021), OGD, SF-OGD (Bhatnagar et al., 2023), decay-OGD (Angelopoulos et al., 2024), PID (Angelopoulos et al., 2023b). A detailed description of existing methods can be found in Appendix C.

**General implement.** We choose target coverage $1 - \alpha = 90\%$ and construct asymmetric prediction sets using two-side quantile scores under $\alpha/2$ respectively. For each baseline, we select the most appropriate range of learning rates $\eta$ for the respective datasets and present the best results in the tables. For sets, all baselines will output asymmetric sets $[\hat{Y}_t - q_t^l, \hat{Y}_t + q_t^u]$ with upper score $q_t^u$ and lower score $q_t^l$ under half of the coverage level $\alpha/2$ respectively.

**Other implement.** For EQ term, we set $f(x) = \frac{1}{1+\exp(-cx)}$ as Sigmoid function and $c = 1$. For ECI-cutoff, we set $h = 1$ and $h_t = h \cdot (\max\{s_{t-w+1}, \ldots, s_t\} - \min\{s_{t-w+1}, \ldots, s_t\})$. For ECI-integral, we set weights $w_i = \frac{0.95^{t-i}}{\sum_{j=1}^{t} 0.95^{t-j}}$ for $1 \leq i \leq t$. Specifically, PID, ECI, and its variants use adaptive learning rates $\eta_t = \eta \cdot (\max\{s_{t-w+1}, \ldots, s_t\} - \min\{s_{t-w+1}, \ldots, s_t\})$, where $w$ is window length. Unless necessary changes are made, the settings for all baselines adhere to the original papers and open-source codes.

**Overview of experimental results.** We have conducted extensive experiments, including stock data in Section 4.2, electricity data in Section 4.3, climate data in Section 4.4, synthetic data in Section 4.5, and ablation study of hyperparameters in Appendix D. Experimental results with Transformer as the base model can be found in Appendix F. More comprehensive experimental results and discussion on the scorecaster and learning rates can be found in Appendix G. Our code is available at `https://github.com/creator-xi/Error-quantified-Conformal-Inference`.

## 4.2 RESULTS IN FINANCE DOMAIN

We consider the uncertainty problem of forecasting stock prices, including Amazon (AMZN) and Google (GOOGL), collected over 9 years (from January 1, 2006 to December 31, 2014). Models will forecast the daily opening price of each of Amazon and Google stock on a log scale.

The quantitative results are shown in Table 1 and Table 2. For ACI, the occurrence of infinite sets is too frequent, due to updating $\alpha_t$ and adopting the $\alpha_t$-quantile of past scores as $q_t$. This implies that ACI may tend to conservatively expand the prediction sets in the face of more complex or volatile data to ensure high coverage rates. However, such a strategy is not always ideal in practical applications, as overly broad sets can reduce the precision and utility of the predictions. For OGD and SF-OGD, they achieve a good balance between coverage rate and set width to some extent. However, their performance overly relies on the selection of learning rate and may fail under many learning rate settings. For PID, its scorecasting can be seen as helping compensate for the base predictors' predictive accuracy. Thus, it can lead to improvements in the worse base predictor case (such as Prophet). However, in the case of better base predictors, it will instead widen the length of the prediction set.

As for ECI, its performance is outstanding, especially in the control of set width. Compared to existing methods, ECI can provide shorter and more accurate prediction sets with little loss of coverage. ECI-cutoff extends ECI by introducing a truncation threshold to further reduce the redundancy of prediction sets. The experimental data also show that ECI-cutoff achieves the shortest set width of all methods in general. ECI-integral extends ECI by integrating the information from longer past time, leading to a better balance between coverage rate and set width.

Figure 3 shows the coverage results on Amazon stock dataset with Prophet as the base predictor. Throughout the entire period, the more effectively a method maintains the nominal level (here is 90%), the more valid the method is. We can see ACI generally having larger oscillations, and OGD,

Table 1: The experimental results in the Amazon stock dataset with nominal level $\alpha = 10\%$. The best (shortest) width results are indicated with bold text.

| Method | Prophet Model | | | AR Model | | | Theta Model | | |
|---|---|---|---|---|---|---|---|---|---|
| | Coverage (%) | Average width | Median width | Coverage (%) | Average width | Median width | Coverage (%) | Average width | Median width |
| ACI | 90.2 | $\infty$ | 46.97 | 89.8 | $\infty$ | 13.77 | 89.7 | $\infty$ | 12.31 |
| OGD | 89.6 | 55.15 | 30.00 | 89.9 | 19.10 | 15.00 | 89.8 | 18.07 | 14.50 |
| SF-OGD | 89.5 | 61.47 | 31.75 | 89.9 | 24.44 | 21.05 | 90.0 | 23.88 | 21.14 |
| decay-OGD | 89.9 | 97.22 | 36.20 | 89.7 | 20.23 | 14.01 | 89.2 | 17.49 | 13.46 |
| PID | 89.8 | 52.56 | 39.09 | 89.6 | 59.22 | 37.93 | 89.5 | 61.19 | 40.20 |
| ECI | 90.1 | 47.00 | 34.84 | 89.5 | 17.12 | 12.73 | 89.7 | 17.46 | 12.49 |
| ECI-cutoff | 89.7 | 43.46 | **29.98** | 89.3 | **16.91** | 12.63 | 89.6 | **17.19** | 12.48 |
| ECI-integral | 89.8 | **42.01** | 30.02 | 89.5 | 16.99 | **12.62** | 89.6 | 17.20 | **12.46** |

Table 2: The experimental results in the Google stock dataset with nominal level $\alpha = 10\%$.

| Method | Prophet Model | | | AR Model | | | Theta Model | | |
|---|---|---|---|---|---|---|---|---|---|
| | Coverage (%) | Average width | Median width | Coverage (%) | Average width | Median width | Coverage (%) | Average width | Median width |
| ACI | 90.0 | $\infty$ | 66.83 | 89.8 | $\infty$ | 18.64 | 90.5 | $\infty$ | 32.78 |
| OGD | 89.7 | 57.60 | 46.00 | 90.7 | 33.76 | 23.00 | 89.9 | 31.49 | 29.50 |
| SF-OGD | 89.6 | 58.92 | 47.78 | 89.9 | 28.31 | 24.42 | 90.0 | 34.04 | 31.48 |
| decay-OGD | 89.9 | 77.23 | 50.18 | 90.2 | 46.53 | 26.77 | 90.2 | 55.32 | 33.71 |
| PID | 90.1 | 57.47 | 48.44 | 89.9 | 64.88 | 54.07 | 89.9 | 63.58 | 54.05 |
| ECI | 89.9 | 56.06 | 46.96 | 89.7 | 19.95 | **17.19** | 89.6 | 30.92 | 29.53 |
| ECI-cutoff | 89.8 | 53.12 | 44.36 | 89.7 | **19.84** | 17.63 | 89.6 | 30.71 | 28.11 |
| ECI-integral | 89.8 | **52.36** | **43.28** | 89.7 | 19.93 | 17.31 | 89.6 | **30.42** | **28.02** |

and SF-OGD become increasingly oscillatory over time. Figure 4 compares the prediction sets of the variants of ECI with those of other methods. Consistent with the quantitative results, the variants of ECI have the shortest prediction sets.

Table 3: The experimental results in the electricity demand dataset with nominal level $\alpha = 10\%$.

| Method | Prophet Model | | | AR Model | | | Theta Model | | |
|---|---|---|---|---|---|---|---|---|---|
| | Coverage (%) | Average width | Median width | Coverage (%) | Average width | Median width | Coverage (%) | Average width | Median width |
| ACI | 90.1 | $\infty$ | 0.443 | 90.1 | $\infty$ | 0.105 | 90.2 | $\infty$ | **0.055** |
| OGD | 89.8 | 0.433 | 0.435 | 90.0 | 0.133 | 0.115 | 90.1 | 0.081 | 0.075 |
| SF-OGD | 89.9 | 0.419 | 0.426 | 90.0 | 0.129 | 0.116 | 90.3 | 0.106 | 0.095 |
| decay-OGD | 90.1 | 0.531 | 0.521 | 90.1 | 0.122 | 0.099 | 90.0 | 0.100 | 0.059 |
| PID | 90.1 | **0.207** | **0.177** | 90.0 | 0.434 | 0.432 | 89.9 | 0.413 | 0.411 |
| ECI | 90.0 | 0.384 | 0.382 | 90.0 | **0.117** | 0.098 | 89.9 | **0.071** | **0.055** |
| ECI-cutoff | 90.0 | 0.405 | 0.396 | 90.2 | 0.118 | **0.096** | 90.1 | 0.072 | **0.055** |
| ECI-integral | 90.1 | 0.402 | 0.398 | 90.0 | **0.117** | 0.098 | 89.9 | 0.072 | **0.055** |

## 4.3 RESULTS IN ENERGY DOMAIN

Then we consider the uncertainty problem of electricity demand. The dataset measures electricity demand in New South Wales, collected at half-hour increments from May 7th, 1996 to December 5th, 1998. All values are normalized in $[0, 1]$.

Table 3 shows that ECI and ECI-integral have the shortest prediction sets with AR and Theta model, even maintaining the highest coverage in the AR model. We can note that PID stands out with Prophet model due to the scorecaster. In fact, This dataset has collected several other variables, such

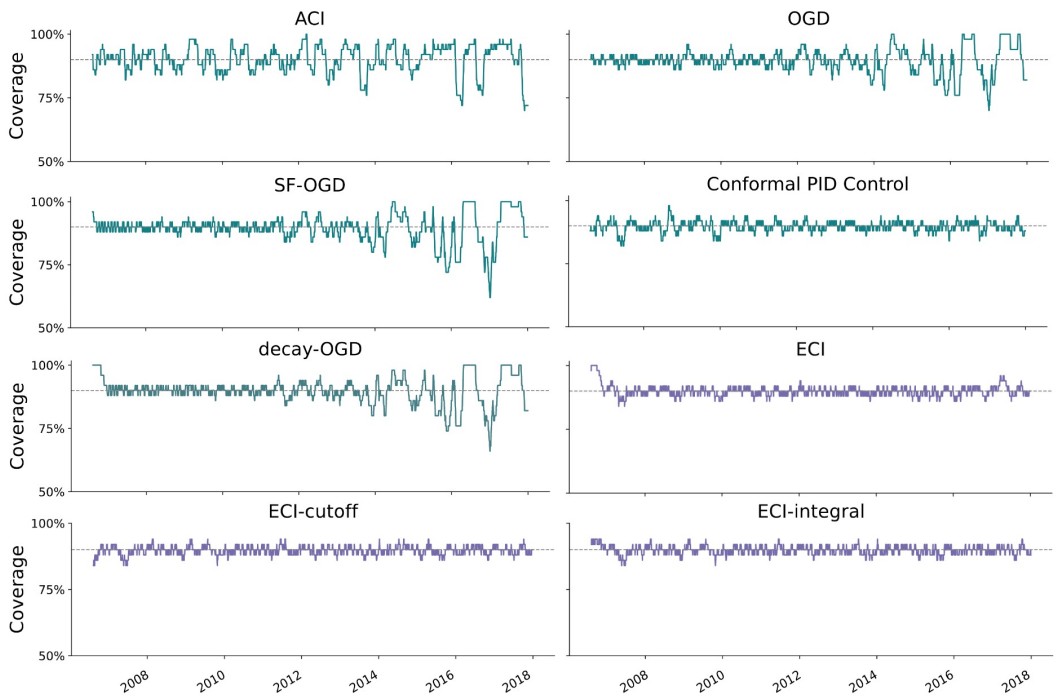

Figure 3: Comparison results of coverage rate on Amazon stock dataset with Prophet model.

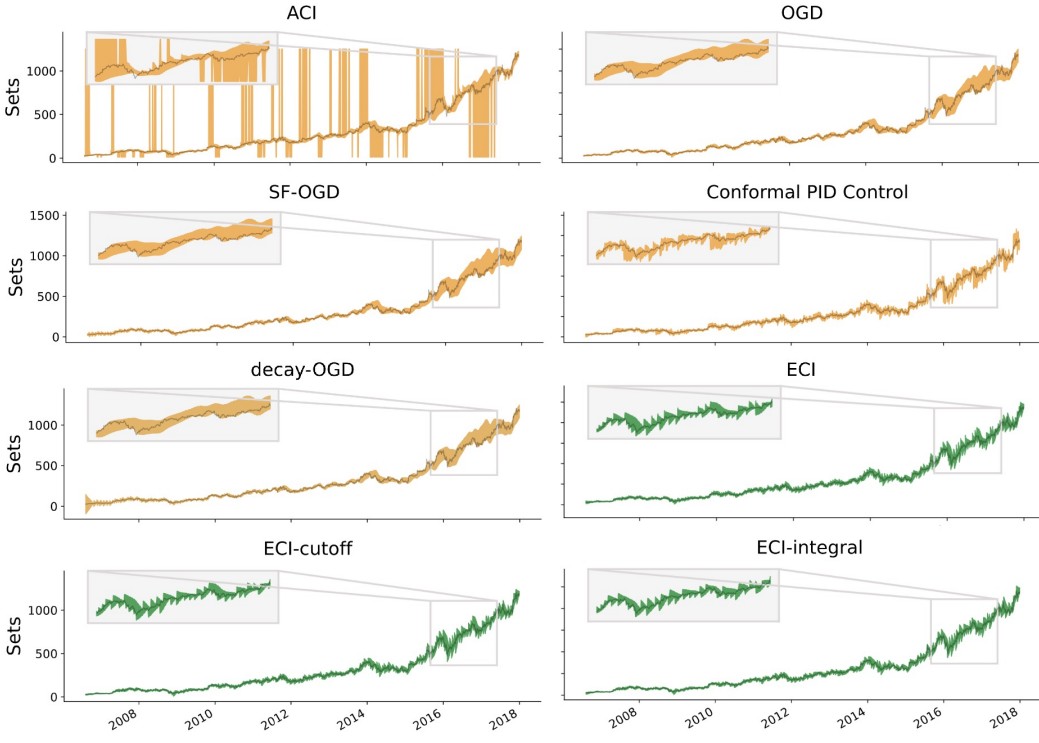

Figure 4: Comparison results of prediction sets on Amazon stock dataset with Prophet model.

as the demand and price in Victoria, the amount of energy transfer between New South Wales and Victoria, and so on. These are given as covariates to the scorecaster and complement Prophet well. Other methods do not use this information.

## 4.4 RESULTS IN CLIMATE DOMAIN

Finally, we consider the uncertainty problem of climate demand. The dataset measures the daily temperature in the city of Delhi over 15 years (from January 1, 2003 to April 24, 2017). Table 4 shows that ECI-cutoff achieves best performance in general.

Table 4: The experimental results in the Delhi temperature dataset with nominal level $\alpha = 10\%$.

| Method | Prophet Model | | | AR Model | | | Theta Model | | |
|---|---|---|---|---|---|---|---|---|---|
| | Coverage (%) | Average width | Median width | Coverage (%) | Average width | Median width | Coverage (%) | Average width | Median width |
| ACI | 91.0 | $\infty$ | 8.49 | 90.0 | $\infty$ | 6.06 | 90.2 | $\infty$ | 6.48 |
| OGD | 90.4 | 7.54 | 7.60 | 90.1 | 6.82 | 6.10 | 90.0 | 6.36 | 6.30 |
| SF-OGD | 90.0 | 7.17 | 7.08 | 90.1 | 6.37 | 5.91 | 90.1 | 6.75 | 6.43 |
| decay-OGD | 90.1 | 8.84 | 8.35 | 90.0 | 6.36 | **5.67** | 89.9 | 6.56 | 6.18 |
| PID | 90.1 | 7.65 | 7.65 | 89.7 | 8.92 | 8.86 | 89.7 | 8.77 | 8.79 |
| ECI | 90.0 | 7.20 | 7.22 | 90.1 | 6.39 | 6.10 | 90.0 | 6.41 | 6.27 |
| ECI-cutoff | 90.1 | **7.01** | **6.96** | 90.1 | **6.28** | 5.97 | 90.0 | **6.27** | **6.17** |
| ECI-integral | 90.0 | 7.21 | 7.30 | 90.2 | 6.39 | 6.11 | 90.0 | 6.38 | 6.26 |

## 4.5 RESULTS IN SYNTHETIC DATA

In this experiment, we compare the performance of our method with other baselines under a synthetic changepoint setting in Barber et al. (2023). The data $\{X_i, Y_i\}_{i=1}^n$ are generated according to a linear model $Y_t = X_t^T \beta_t + \epsilon_t$, $X_t \sim \mathcal{N}(0, I_4)$, $\epsilon_t \sim \mathcal{N}(0, 1)$. And we set: $\beta_t = \beta^{(0)} = (2, 1, 0, 0)^\top$ for $t = 1, \ldots, 500$; $\beta_t = \beta^{(1)} = (0, -2, -1, 0)^\top$ for $t = 501, \ldots, 1500$; and $\beta_t = \beta^{(2)} = (0, 0, 2, 1)^\top$ for $t = 1501, \ldots, 2000$. And two changes in the coefficients happen up to time 2000.

We compare ECI and its variants with some competing methods about the coverage and prediction set width. Table 5 show the result of coverage and set width, while the base predictor is Prophet model. In general, ECI-cutoff and ECI-integral achieve best performance.

Table 5: The experimental results in the synthetic data dataset with nominal level $\alpha = 10\%$.

| Method | Prophet Model | | | AR Model | | | Theta Model | | |
|---|---|---|---|---|---|---|---|---|---|
| | Coverage (%) | Average width | Median width | Coverage (%) | Average width | Median width | Coverage (%) | Average width | Median width |
| ACI | 89.9 | $\infty$ | 8.20 | 89.9 | $\infty$ | 8.20 | 89.9 | $\infty$ | 8.43 |
| OGD | 90.0 | 8.49 | 8.50 | 89.9 | 8.39 | 8.40 | 89.9 | 8.73 | 8.70 |
| SF-OGD | 90.0 | 12.48 | 11.56 | 90.0 | 12.58 | 11.69 | 89.9 | 12.70 | 11.88 |
| decay-OGD | 90.0 | 8.30 | **8.22** | 90.0 | 8.26 | 8.21 | 90.0 | 8.57 | 8.60 |
| PID | 89.7 | 11.02 | 9.64 | 89.9 | 10.83 | 9.35 | 89.7 | 11.23 | 9.78 |
| ECI | 89.9 | **8.16** | 8.25 | 89.9 | 8.17 | 8.26 | 89.8 | 8.55 | 8.68 |
| ECI-cutoff | 89.8 | 8.31 | 8.44 | 89.9 | **8.14** | **8.19** | 89.8 | 8.51 | 8.59 |
| ECI-integral | 89.8 | 8.25 | 8.37 | 89.9 | 8.16 | 8.23 | 89.8 | **8.48** | **8.58** |

## 5 CONCLUSION

Several approaches have recently been introduced for online conformal inference. A significant limitation of these methods is their neglect of quantifying extent of over/under coverage. In this work, we propose *Error-quantified Conformal Inference* (ECI) to construct prediction sets for time series data. Compared with ACI (Gibbs & Candès, 2021) and its variants, ECI introduces additional smooth feedback by measuring the magnitude of error $s_t - q_t$. ECI can rapidly adapt to the distributional shifts in time series, and yield more tight conformal prediction sets. Theoretically, we establish a finite-sample coverage guarantee for ECI with a fixed learning rate in a short interval and prove a miscoverage bound with arbitrary learning rate. Empirically, we verify our method's effectiveness and efficiency across extensive datasets.

ACKNOWLEDGEMENT

We would like to thank the anonymous reviewers and area chair for their helpful comments. Changliang Zou was supported by the National Key R&D Program of China (Grant Nos. 2022YFA1003703, 2022YFA1003800), and the National Natural Science Foundation of China (Grant Nos. 11925106, 12231011, 12326325).

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

# A SMOOTHED WAYS

## A.1 EXPLANATIONS FOR EXTRA BIAS

Our target is the quantile loss $\ell_t(q) = (\text{err}_t - \alpha)(s_t - q)$. For simplicity, we denote $h(q) = \text{err}_t - \alpha, g(q) = s_t - q$. Note that $h(q)$ is non-differentiable, so we use smoothed function $f(q)$ to approximate $h(q)$. The fully smoothed method in Equation (4) considers:

$$\nabla h(q)g(q) \approx \nabla f(q)g(q)$$
$$= f(q)\nabla g(q) + g(q)\nabla f(q).$$

The two terms $f(q)\nabla g(q)$ and $g(q)\nabla f(q)$ both introduce error. Strictly we have

$$\nabla h(q)g(q) = \lim_{\delta \to 0} \frac{h(q+\delta)g(q+\delta) - h(q+\delta)g(q) + h(q+\delta)g(q) - h(q)g(q)}{\delta}$$
$$= \lim_{\delta \to 0} \frac{h(q+\delta)g(q+\delta) - h(q+\delta)g(q)}{\delta} + \lim_{\delta \to 0} \frac{h(q+\delta)g(q) - h(q)g(q)}{\delta}$$
$$= h(q)\nabla g(q) + \underbrace{\lim_{\delta \to 0} \frac{h(q+\delta) - h(q)}{\delta}}_{\text{using } \nabla f \text{ to approximate}} g(q)$$
$$\approx -(\text{err}_t - \alpha) - \nabla f(s_t - q)(s_t - q).$$

Hence, if we only approximate the second term, potential bias may be reduced. The fully smoothed method is better only when the error of the two terms are negatively correlated and cancel out.

## A.2 FULLY SMOOTHED WAYS

There is another smoothing technique applied in the quantile regression (Fernandes et al., 2021; Tan et al., 2022), which directly smooths the inidcator in subgradient. Based on this, we can have the following udpate rule

$$q_{t+1} = q_t + \eta \cdot \left[ f(s_t - q_t) - \alpha \right]. \tag{10}$$

To further validate our ideas in Section 3.1, we test the experimental performance of smoothed method and fully smoothed method, as shown in Figures 5 to 7. All experimental setting are aligned with Section 4.

The update rule of fully smoothed method in Equation (4) is

$$q_{t+1} = q_t + \eta \cdot \left[ f(s_t - q_t) - \alpha + (s_t - q_t)\nabla f(s_t - q_t) \right].$$

And the update rule of ECI in Equation (5) is

$$q_{t+1} = q_t + \eta \cdot \left[ \text{err}_t - \alpha + (s_t - q_t)\nabla f(s_t - q_t) \right].$$

Compared with ECI in Equation (5), fully smoothed updates rules in Equation (4) and Equation (10) do not keep the actual value of indicator function $\text{err}_t$, and brings the bias between smooth function $f(s_t - q_t)$ and $\mathbb{1}(s_t > q_t)$. This leads to these method being conservative.

Experimental results also demonstrate it. Since $f(x)$ does not approach $0$ quickly in the early part of the period, and may even be larger than $\alpha$, the coverage rate can not effectively approach $1 - \alpha$. It can be seen that smoothed method and fully smoothed method tends to have overly high coverage in the early stages in the Figures 5 to 7. Consequently, it results in wider sets.

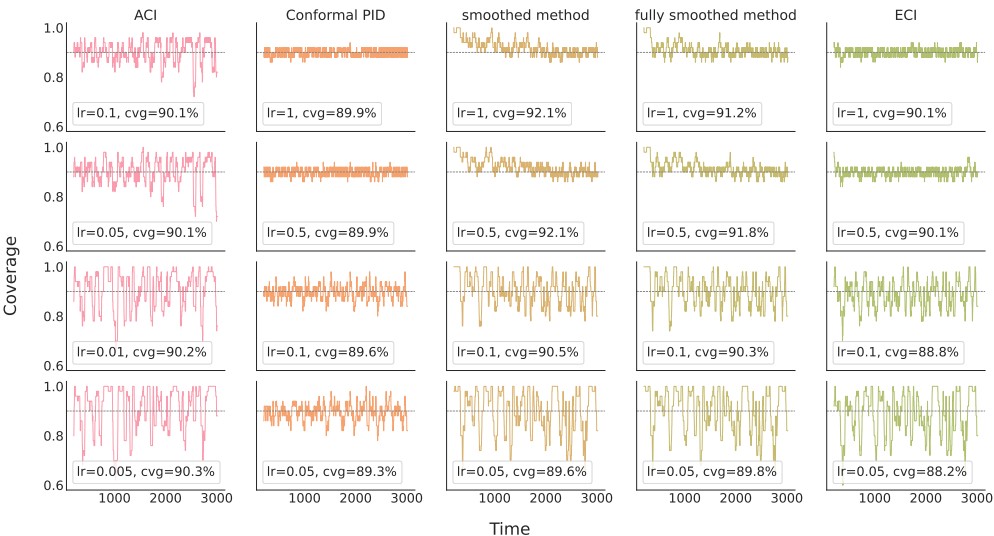

Figure 5: Coverage result on Amazon stock dataset with Prophet model.

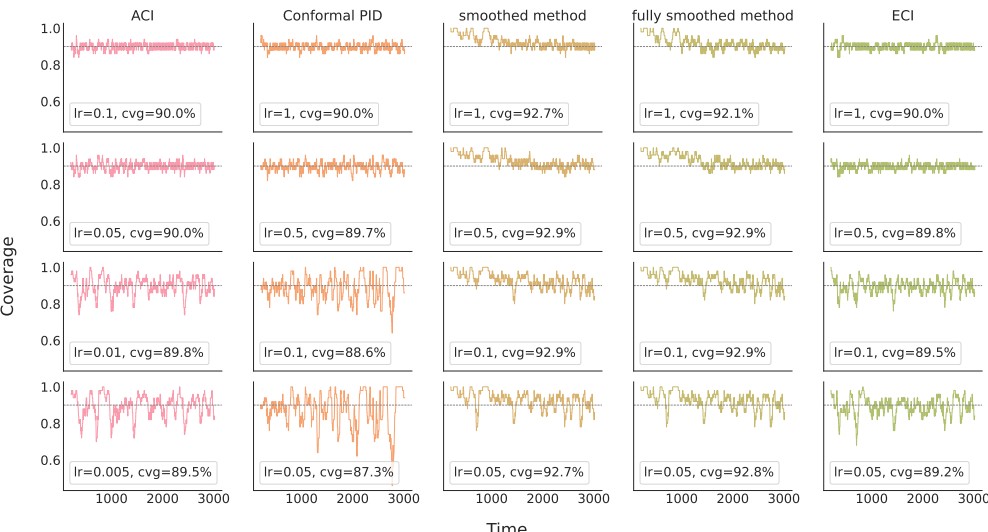

Figure 6: Coverage result on Amazon stock dataset with AR model.

# B  PROOFS OF MAIN RESULTS

We show below that under Assumptions 1 and 2, there exists a bound for $q_t$ and EQ term, depending on learning rate $\eta_t$ and the upper bound of $s_t$. This result is essential in proving our following results of coverage guarantees.

## B.1  PROPOSITIONS

**Proposition 1.** *Fix an initial threshold* $q_1 \in [0, B]$. *Then under Assumptions 1 and 2, ECI in* (5) *with arbitrary nonnegative learning rate* $\eta_t$ *satisfies that*

$$-(\alpha + \lambda)M_{t-1} \leq q_t \leq B + (1 - \alpha + \lambda)M_{t-1} \quad \forall t \geq 1, \tag{11}$$

*where* $M_0 = 0$, *and* $M_t = \max_{1 \leq r \leq t} \eta_r$ *for* $t \geq 1$.

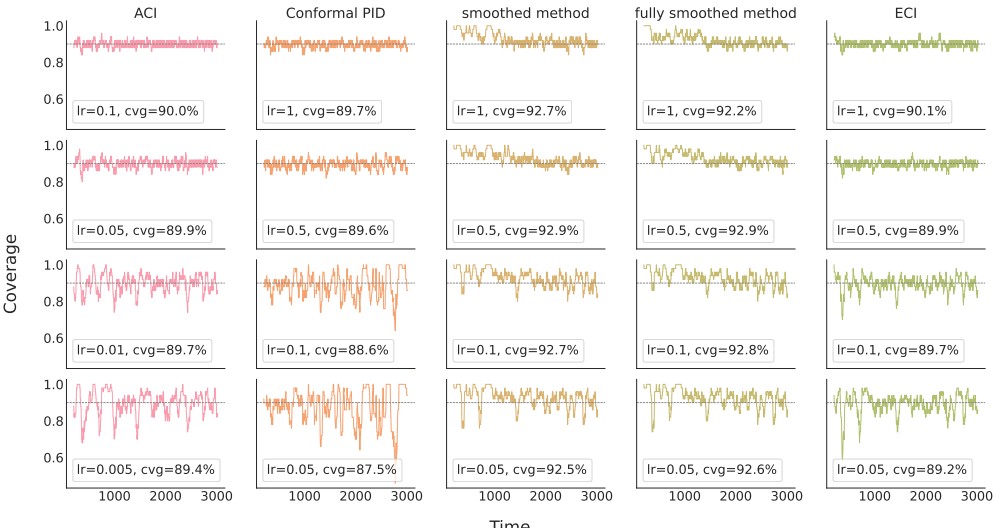

Figure 7: Coverage result on Amazon stock dataset with Theta model.

*Proof.* We prove this by induction. First, $q_1 \in [0, B]$ by assumption. Next fix any $t \geq 1$ and assume $q_t$ lies in the range specified in (11), and consider $q_{t+1}$.

(1): $s_t \geq q_t$,

$$
\begin{aligned}
q_{t+1} &= q_t + \eta_t (1 - \alpha) + \eta_t (s_t - q_t) \, \nabla f(s_t - q_t) \\
&\leq q_t + \eta_t (1 - \alpha) + \eta_t \lambda \\
&\leq s_t + (1 - \alpha + \lambda) \eta_t \\
&\leq B + (1 - \alpha + \lambda) M_t
\end{aligned}
$$

and $q_{t+1} \geq q_t \geq -M_{t-1} \alpha \geq -M_t \alpha$.

(2): $s_t \leq q_t$,

$$
\begin{aligned}
q_{t+1} &= q_t + \eta_t (-\alpha) + \eta_t (s_t - q_t) \, \nabla f(s_t - q_t) \\
&\geq q_t + \eta_t (-\alpha) - \eta_t \lambda \\
&\geq -(\alpha + \lambda) M_t,
\end{aligned}
$$

and $q_{t+1} \leq q_t \leq B + (1 - \alpha + \lambda) M_{t-1} \leq B + (1 - \alpha + \lambda) M_t$. $\qquad\square$

**Proposition 2.** *Under Assumptions 1 and 2, $|(s_t - q_t)\nabla f(s_t - q_t)| \leq c\,[B + (1 - \alpha + \lambda) M_{t-1}]$ for any $t \geq 1$.*

*Proof.* Based on $s_t \in [0, B]$ and Proposition 1,

$$
\begin{aligned}
|s_t - q_t| &= \max\{q_t - s_t, s_t - q_t\} \leq \max\{q_t, s_t - q_t\} \\
&\leq \max\{B + (1 - \alpha + \lambda) M_{t-1}, B + (\alpha + \lambda) M_{t-1}\} \\
&\leq B + (1 - \alpha + \lambda) M_{t-1}.
\end{aligned}
$$

Hence $|(s_t - q_t)\nabla f(s_t - q_t)| \leq c\,[B + (1 - \alpha + \lambda) M_{t-1}]$ $\qquad\square$

### B.2 PROOF OF THEOREM 1

**Theorem 1.** Assume that $\eta > 2NB$, $c < \frac{\min\{\eta, N^2\}}{2N^2[B + (1 - \alpha + \lambda)\eta]}$, where $N = \lceil \frac{1}{\alpha} \rceil$. Under Assumptions 1 and 2, the prediction set generated by eq. (5) satisfies:

$$
\lim_{T \to \infty} \frac{1}{T} \sum_{t=1}^{T} \mathbb{1}\{Y_t \notin \hat{C}_t\} \leq \alpha. \tag{12}
$$

*Proof.* We first prove that for any $t$, $s_t > q_t$ implies $s_{t+i} < q_{t+i}$, $i = 1, 2, \cdots, N-1$. Note that $\{Y_t \in \hat{C}_t\} = \{|Y_t - \hat{Y}_t| \leq q_t\}$ is meaningless when $q_t < 0$. Hence we set $q_t$ to be $\max\{q_t, 0\}$ after each update ( which does not affect the validity of our proof ) and assume $q_t \geq 0$. For simplicity, denote $g(x) = x\nabla f(x)$, then $q_{t+1} = q_t + \eta\left[\text{err}_t - \alpha + g(s_t - q_t)\right]$.

We prove by induction. For $k = 1$,

$$q_{t+1} - s_{t+1} = q_t + \eta\left[1 - \alpha + g(s_t - q_t)\right] > \eta(1 - \alpha - c\left[B + (1 - \alpha + \lambda)\eta\right])$$

Observe that $c \cdot \left[B + (1 - \alpha + \lambda)\eta\right] < \left[B + (1 - \alpha + \lambda)\eta\right] \cdot \frac{N^2}{2N^2[B+(1-\alpha+\lambda)\eta]} = \frac{1}{2}$, hence

$$q_{t+1} - s_{t+1} > \eta(1 - \alpha - \frac{1}{2}) > \eta(\frac{1}{2} - \alpha) \geq 0.$$

For $2 \leq k \leq N-1$, by recursion:

$$
\begin{aligned}
q_{t+k} - s_{t+k} &= q_t + \eta(1 - k\alpha) + \eta\sum_{i=0}^{k-1} g(s_{t+i} - q_{t+i}) - s_{t+k} \\
&> \eta(1 - k\alpha) + \eta\sum_{i=1}^{k-1} g(s_{t+i} - q_{t+i}) - s_{t+k} \qquad (s_t > q_t \geq 0) \\
&\geq \eta(1 - k\alpha) + c\eta\sum_{i=1}^{k-1}(s_{t+i} - q_{t+i}) - s_{t+k} \\
&\geq \eta(1 - k\alpha) - c(k-1)\left[B + (1 - \alpha + \lambda)\eta\right] - s_{t+k} \\
&\geq \frac{\eta}{N} - c(N-2)\left[B + (1 - \alpha + \lambda)\eta\right] - s_{t+k} \qquad (k \leq N-1, \ \alpha \geq \frac{1}{N}) \\
&> \frac{\eta}{N} - \frac{(N-2)\eta}{2N^2} - \frac{\eta}{2N} > 0
\end{aligned}
$$

The last inequality is based on the assumption $s_{t+k} \leq B < \frac{\eta}{2N}, c < \frac{\eta}{2N^2[B+(1-\alpha+\lambda)\eta]}$.

In conclusion, we have proved that for every miscoverage step $t$, i.e. $Y_t \notin \hat{C}_t$, the next $N-1$ steps of (5) will satisfy $Y_{t+i} \in \hat{C}_{t+i}, i = 1, 2, \cdots N-1$. Therefore, for any $T$,

$$\frac{1}{N}\sum_{t=T+1}^{T+N} \mathbb{1}\{Y_t \notin \hat{C}_t\} \leq \frac{1}{N}.$$

$\square$

### B.3 PROOF OF THEOREM 2

**Theorem 2.** Let $\{\eta_t\}_{t \geq 1}$ be an arbitrary positive sequence. Under Assumptions 1 and 2, the prediction set generated by Equation (5) with adaptive learning rate $\eta_t$ satisfies:

$$\left|\frac{1}{T}\sum_{t=1}^{T}(\text{err}_t - \alpha)\right| \leq \frac{(B + M_{T-1})\|\Delta_{1:T}\|_1}{T} + c\left[B + (1 - \alpha + \lambda)M_{T-1}\right] \tag{13}$$

where $\|\Delta_{1:T}\|_1 = |\eta_1^{-1}| + \sum_{t=2}^{T}|\eta_t^{-1} - \eta_{t-1}^{-1}|, M_T = \max_{1 \leq r \leq T}\eta_r$.

*Proof.* Denote $\Delta_1 = \eta_1^{-1}$, and $\Delta_t = \eta_t^{-1} - \eta_{t-1}^{-1}$ for all $t \geq 1$,

$$
\left| \frac{1}{T} \sum_{t=1}^{T} (\text{err}_t - \alpha) \right| = \left| \frac{1}{T} \sum_{t=1}^{T} \left( \sum_{r=1}^{t} \Delta_r \right) \cdot \eta_t \left( \text{err}_t - \alpha \right) \right|
$$

$$
= \left| \frac{1}{T} \sum_{r=1}^{T} \Delta_r \left( \sum_{t=r}^{T} \eta_t \left( \text{err}_t - \alpha \right) \right) \right|
$$

$$
= \left| \frac{1}{T} \sum_{r=1}^{T} \Delta_r \left( q_{T+1} - q_r + \sum_{t=r}^{T} \eta_t (s_t - q_t) \nabla f(s_t - q_t) \right) \right| \quad \text{by Equation (5)}
$$

$$
\leq \frac{1}{T} \left| \sum_{r=1}^{T} \Delta_r (q_{T+1} - q_r) \right| + \frac{1}{T} \left| \sum_{r=1}^{T} \Delta_r \sum_{t=r}^{T} \eta_t (s_t - q_t) \nabla f(s_t - q_t) \right|
$$

$$
\leq \frac{1}{T} \left| \sum_{r=1}^{T} \Delta_r (q_{T+1} - q_r) \right| + \frac{1}{T} \left| \sum_{t=1}^{T} (s_t - q_t) \nabla f(s_t - q_t) \right|
$$

$$
\leq \frac{1}{T} \|\Delta_{1:T}\|_1 \max_{1 \leq r \leq T} |q_{T+1} - q_r| + \frac{1}{T} \left( \sum_{t=1}^{T} |s_t - q_t| \nabla f(s_t - q_t) \right)
$$

$$
\leq \frac{(B + M_{T-1})\|\Delta_{1:T}\|_1}{T} + c \cdot \frac{\sum_{t=1}^{T} [B + (1 - \alpha + \lambda) M_{t-1}]}{T}
$$

$$
\leq \frac{(B + M_{T-1})\|\Delta_{1:T}\|_1}{T} + c \left[ B + (1 - \alpha + \lambda) M_{T-1} \right].
$$

$\square$

## C   MORE DETAILS ON EXISTING METHODS

### C.1   ACI

Adaptive Conformal Inference (ACI) in Algorithm 1 models the sequentially conformal inference with distribution shift as a learning problem of a single parameter whose optimal value is varying over time. Assume we have a calibration set $\mathcal{D}_{\text{cal}} \subseteq \{(X_r, Y_r)\}_{1 \leq r \leq t-1}$, and $\hat{Q}_t(\cdot)$ is the fitted quantiles of the non-conformity scores:

$$
\hat{Q}(p) := \inf \left\{ s : \left( \frac{1}{|\mathcal{D}_{\text{cal}}|} \sum_{(X_r, Y_r) \in \mathcal{D}_{\text{cal}}} \mathbb{1}_{\{S(X_r, Y_r) \leq s\}} \right) \geq p \right\}.
$$

For prediction set $\hat{C}_t(\alpha) := \{y : S_t(X_t, y) \leq \hat{Q}_t(1 - \alpha)\}$, define:

$$
\beta_t := \sup\{\beta : Y_t \in \hat{C}_t(\beta)\}.
$$

Consider pinball loss $\ell(\alpha_t, \beta_t) = \rho_\alpha(\beta_t - \alpha_t)$, by gradient descent:

$$
\alpha_{t+1} = \alpha_t - \eta \partial_{\alpha_t} \ell(\alpha_t, \beta_t) = \alpha_t + \eta(\alpha - \mathbb{1}_{\alpha_t > \beta_t}) = \alpha_t + \eta(\alpha - \text{err}_t).
$$

ACI transforms unbounded score sequences into bounded ones, which then implies long-run coverage for any score sequence. This may, however, come at a cost: ACI can sometimes output infinite or null prediction sets ($\alpha_t < 0$ or $\alpha_t > 1$).

### C.2   OGD

Online Gradient Descent (OGD) in Algorithm 2 is an iterative optimization algorithm that updates model parameters incrementally using each new data point, making it suitable for real-time and streaming data applications.

---

**Algorithm 1** Adaptive Conformal Inference (ACI)

---

**Require:** $\alpha \in (0,1)$, $\eta > 0$, $D_{cal}$, init. $\alpha_1 \in \mathbb{R}$

1: **for** $t \geq 1$ **do**
2:     Observe input $X_t \in \mathcal{X}$
3:     Compute $\hat{Q}_t(1 - \alpha_t)$
4:     Return prediction set $\hat{C}_t(\alpha_t) := \{y : S_t(X_t, y) \leq \hat{Q}_t(1 - \alpha_t)\}$
5:     Observe true label $Y_t \in \mathcal{Y}$ and compute true radius $\beta_t := \sup\{\beta : Y_t \in \hat{C}_t(\beta)\}$
6:     Update predicted radius
$$\alpha_{t+1} = \alpha_t + \eta(\alpha - \mathbb{1}_{\alpha_t > \beta_t})$$
7: **end for**

---

**Algorithm 2** Online Gradient Descent (OGD)

---

**Require:** $\alpha \in (0,1)$, base predictor $\hat{f}$, learning rate $\eta > 0$, init. $q_1 \in \mathbb{R}$

1: **for** $t \geq 1$ **do**
2:     Observe input $X_t \in \mathcal{X}$
3:     Return prediction set $\hat{C}_t(X_t, q_t) = [\hat{f}_t(X_t) - q_t, \hat{f}_t(X_t) + q_t]$
4:     Observe true label $Y_t \in \mathcal{Y}$ and compute true radius $s_t = \inf\{s \in \mathbb{R} : Y_t \in \hat{C}_t(X_t, s)\}$
5:     Compute quantile loss $\ell^{(t)}(q_t) = \rho_{1-\alpha}(s_t - q_t)$
6:     Update predicted radius
$$q_{t+1} = q_t - \eta \nabla \ell^{(t)}(q_t)$$
7: **end for**

---

## C.3   SF-OGD

Scale-Free OGD (SF-OGD) that we summarize in Algorithm 3 is a variant of OGD that decays its effective learning rate based on cumulative past gradient norms. Suppose $\hat{f}$ is a base predictor and choose $\widehat{C}_t(X_t, q) := [\hat{f}_t(X_t) - q, \hat{f}_t(X_t) + q]$ to be a prediction set around $\hat{f}_t(X_t)$.

---

**Algorithm 3** Scale-Free Online Gradient Descent (SF-OGD)

---

**Require:** $\alpha \in (0,1)$, base predictor $\hat{f}$, learning rate $\eta > 0$, init. $q_1 \in \mathbb{R}$

1: **for** $t \geq 1$ **do**
2:     Observe input $X_{t+1} \in \mathcal{X}$
3:     Return prediction set $\hat{C}_t(X_t, q_t) = [\hat{f}_t(X_t) - q_{t+1}, \hat{f}_t(X_t) + q_t]$
4:     Observe true label $Y_t \in \mathcal{Y}$
5:     Compute true radius $s_t = \inf\{s \in \mathbb{R} : Y_t \in \hat{C}_t(X_t, s)\}$
6:     Compute quantile loss $\ell^{(t)}(q_t) = \rho_{1-\alpha}(s_t - q_t)$
7:     Update predicted radius

$$q_{t+1} = q_t - \eta \frac{\nabla \ell^{(t)}(q_t)}{\sqrt{\sum_{i=1}^{t} \|\nabla \ell^{(i)}(q_i)\|_2^2}}$$

8: **end for**

---

## C.4   DECAY-OGD

Online conformal prediction with decaying step sizes (decay-OGD) in Algorithm 4 is a variant of OGD that decays its effective learning rate based on time steps. Suppose $\hat{f}$ is a base predictor and choose $\widehat{C}_t(X_t, q) := [\hat{f}_t(X_t) - q, \hat{f}_t(X_t) + q]$ to be a prediction set around $\hat{f}_t(X_t)$.

---

**Algorithm 4** Online conformal prediction with decaying step sizes (decay-OGD)

---

**Require:** $\alpha \in (0, 1)$, base predictor $\hat{f}$, learning rate $\eta > 0$, init. $q_1 \in \mathbb{R}$

1: **for** $t \geq 1$ **do**
2:     Observe input $X_{t+1} \in \mathcal{X}$
3:     Return prediction set $\hat{C}_t(X_t, q_t) = [\hat{f}_t(X_t) - q_{t+1}, \hat{f}_t(X_t) + q_t]$
4:     Observe true label $Y_t \in \mathcal{Y}$
5:     Compute true radius $s_t = \inf\{s \in \mathbb{R} : Y_t \in \hat{C}_t(X_t, s)\}$
6:     Compute quantile loss $\ell^{(t)}(q_t) = \rho_{1-\alpha}(s_t - q_t)$
7:     Compute a decaying step size $\eta_t = \eta \cdot t^{-\frac{1}{2}-\epsilon}$
8:     Update predicted radius
$$q_{t+1} = q_t - \eta_t \nabla \ell^{(t)}(q_t)$$
9: **end for**

---

## C.5 Conformal PID

Conformal PID in Algorithm 5 is bulit upon ideas from conformal prediction and control theory. It is able to prospectively model conformal scores in an online setting, and adapt to the presence of systematic errors due to seasonality, trends, and general distribution shifts. Conformal PID consists of three parts: quantile tracking, error integration and scorecasting. For prediction set:

$$\mathcal{C}_t = \{y \in \mathcal{Y} : S_t(x_t, y) \leq q_t\},$$

consider the optimization:

$$\underset{q \in \mathbb{R}}{\text{minimize}} \sum_{t=1}^{T} \rho_{1-\alpha}(s_t - q).$$

Conformal PID solves it via online gradient method:

$$
\begin{aligned}
q_{t+1} &= q_t + \eta \partial \rho_{1-\alpha}(s_t - q_t) \\
&= q_t + \eta(\mathbb{1}(s_t > q_t) - \alpha) = q_t + \eta(\text{err}_t - \alpha),
\end{aligned}
$$

which is called quantile tracking. In parctice, the learning rate is not fixed. They choose it in an adaptive way: $\eta_t = \eta \cdot (\max\{s_{t-w+1}, \cdots, s_t\} - \min\{s_{t-w+1}, \cdots, s_t\})$, where $\eta$ is a scale parameter. The error integration incorporates the past error to further stabilize the coverage :

$$q_{t+1} = r_t\left(\sum_{i=1}^{t}(\text{err}_i - \alpha)\right).$$

The last step is to add up a scorecasting term: $g_t^{'}$, a model that can take advantage of any leftover signal that is not captured like seasonality, trends, and exogenous covariates. Scorecastor needs be trained and can be Theta or other models.

Putting the three steps together is the conformal PID method:

$$q_{t+1} = g_t' + \eta_t(\text{err}_t - \alpha) + r_t\left(\sum_{i=1}^{t}(\text{err}_i - \alpha)\right)$$

## C.6 SPCI

The sequential predictive conformal inference (SPCI) outlined in Algorithm 6 cast the conformal prediction set as predicting the quantile of a future residual and adaptively re-estimate the conditional quantile of non-conformity scores. Suppose $\hat{f}$ is a pre-trained model, $\widehat{Q}_t(p)$ is an estimator of $Q_t(p)$, the $p-$th quantile of the residual $\hat{\epsilon}_t = |Y_t - \hat{f}(X_t)|$. SPCI sets is $\hat{C}_{t-1}(X_t)$ is defined as:

$$[\hat{f}(X_t) + \widehat{Q}_t(\hat{\beta}), \hat{f}(X_t) + \widehat{Q}_t(1 - \alpha + \hat{\beta})],$$

---

**Algorithm 5** Conformal PID

---

**Require:** $\alpha \in (0,1)$, base predictor $\hat{f}$, trained scorecastor $g'$, $\eta > 0$, window length $w$, init. $q_1 \in \mathbb{R}$
1: **for** $t \geq 1$ **do**
2:      Observe input $X_t \in \mathcal{X}$
3:      Return prediction set $\mathcal{C}_t = \{y \in \mathcal{Y} : S_t(x_t, y) \leq q_t\}$
4:      Observe true label $Y_t \in \mathcal{Y}$ and compute score $s_t := S_t(X_t, Y_t)$,
5:      Compute learning rate $\eta_t = \eta \cdot (\max\{s_{t-w+1}, \cdots, s_t\} - \max\{s_{t-w+1}, \cdots, s_t\})$
6:      Compute the integrator

$$r_t = r_t\bigg( \sum_{i=1}^{t} (\mathrm{err}_i - \alpha) \bigg)$$

7:      Compute the scorecastor $g'_t(X_t)$
8:      Update :

$$q_{t+1} = g'_t(X_t) + \eta_t(\mathrm{err}_t - \alpha) + r_t\bigg( \sum_{i=1}^{t} (\mathrm{err}_i - \alpha) \bigg)$$

9: **end for**

---

where $\hat{\beta}$ minimizes set width:

$$\hat{\beta} = \arg \min_{\beta \in [0, \alpha]} (\widehat{Q}_t(1 - \alpha + \beta) - \widehat{Q}_t(\beta)).$$

---

**Algorithm 6** Sequential Predictive Conformal Inference (SPCI)

---

**Require:** Training data $\{(X_t, Y_t)\}_{t=1}^T$, prediction algorithm $\mathcal{A}$, significance level $\alpha$, quantile regression algorithm $\mathcal{Q}$.
**Output:** Prediction sets $\hat{C}_{t-1}(X_t) = [\hat{f}(X_t) + \widehat{Q}_t(\hat{\beta}), \hat{f}(X_t) + \widehat{Q}_t(1 - \alpha + \hat{\beta})], t > T$
1: Obtain $\hat{f}$ and prediction residuals $\hat{\varepsilon}$ with $\mathcal{A}$ and $\{(X_t, Y_t)\}_{t=1}^T$
2: **for** $t > T$ **do**
3:      Use quantile regression to obtain $\hat{Q}_t \leftarrow \mathcal{Q}(\hat{\varepsilon})$
4:      Obtain prediction set $\hat{C}_{t-1}(X_t)$
5:      Obtain new residual $\hat{\varepsilon}_t$
6:      Update residuals $\hat{\varepsilon}$ by sliding one index forward (i.e., add $\hat{\varepsilon}_t$ and remove the oldest one)
7: **end for**

---

# D    ABLATION STUDY OF HYPERPARAMETERS

## D.1    EFFECTS OF DIFFERENT SCALE PARAMETER IN SIGMOID FUNCTION

We explore the effects of different scale parameter $c$ in Sigmoid function. We conduct the ablation study on Amazon stock dataset. Figures 8 and 9 show the line graphs across different $c$. When base predictor is not well (such as Prophet), as $c$ increases, the sets tighten, but the coverage decreases. For AR model and Theta model, there are almost identical performance when $c$ is in a reasonable range. Note that, when $c$ is large, numeric overflow will encounter because of scalar power. As $c$ varies, the changes in coverage and width are relatively small and the two metrics are actually a trade-off. When the coverage is fixed, the width of our methods with $c \in \{0.1, 0.5, 1, 1.5, 2\}$ are consistently shorter than other methods (see Table 1). In general, our methods are less sensitive to the choice of scale parameter $c$ due to the trade-off between coverage and width.

## D.2    EFFECTS OF DIFFERENT WINDOW LENGTH

We also explore the effects of different window length $w$ in adaptive learning rates $\eta_t$ and adaptive cutoff threshold $h_t$. Tables 6 to 8 show the experimental results of window length $w$ in ECI-cutoff. In general, $w = 100$ has the shortest average width with coverage greater than 89.5%. It is worth noting that in the synthetic dataset, there is a clear trend of increasing width and coverage as $w$ increases.

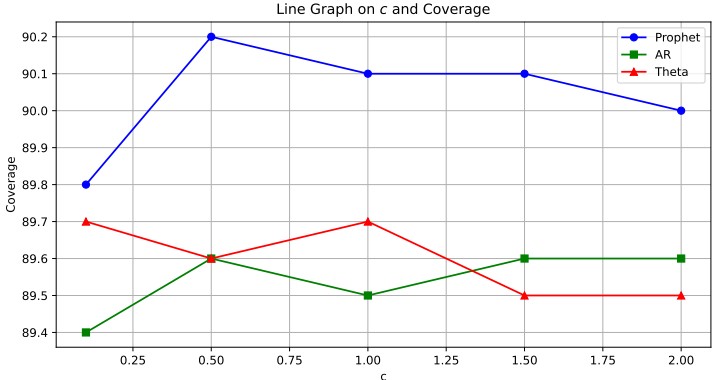

Figure 8: Coverage result on different scale parameter $c$ in Sigmoid function.

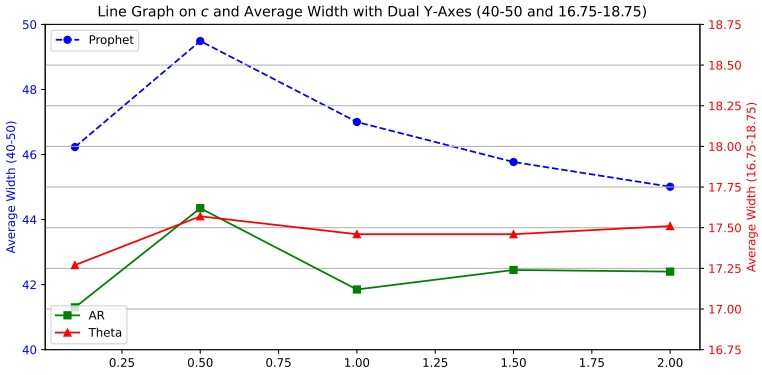

Figure 9: Set width result on different scale parameter $c$ in Sigmoid function.

This is because the main influencing factor at this time is the learning rate, and an increase in $w$ leads to a larger adaptive learning rate.

Table 6: The ablation experimental results of window length $w$ in the synthetic data dataset with nominal level $\alpha = 10\%$.

| $w$ | Prophet Model | | | AR Model | | | Theta Model | | |
|---|---|---|---|---|---|---|---|---|---|
| | Coverage ( %) | Average width | Median width | Coverage ( %) | Average width | Median width | Coverage ( %) | Average width | Median width |
| 10 | 89.4 | 8.18 | 8.21 | 89.4 | 8.22 | 8.30 | 89.4 | 8.41 | 8.43 |
| 50 | 89.8 | 8.42 | 8.45 | 89.6 | 8.32 | 8.37 | 89.6 | 8.61 | 8.57 |
| 100 | 89.8 | 8.31 | 8.44 | 89.9 | 8.14 | 8.19 | 89.8 | 8.51 | 8.59 |
| 150 | 89.9 | 8.47 | 8.46 | 89.9 | 8.43 | 8.47 | 89.9 | 8.83 | 8.82 |
| 200 | 89.9 | 8.56 | 8.54 | 89.9 | 8.33 | 8.40 | 89.9 | 8.83 | 8.84 |

# E    EXPERIMENTAL RESULTS IN SYNTHETIC DATA

Following Barber et al. (2023), we test the performance on synthetic data under a changepoint setting. The data $\{X_i, Y_i\}_{i=1}^n$ are generated according to a linear model $Y_t = X_t^T \beta_t + \epsilon_t$, $X_t \sim$

Table 7: The ablation experimental results of window length $w$ in the Amazon stock dataset with nominal level $\alpha = 10\%$.

| $w$ | Prophet Model | | | AR Model | | | Theta Model | | |
|---|---|---|---|---|---|---|---|---|---|
| | Coverage (%) | Average width | Median width | Coverage (%) | Average width | Median width | Coverage (%) | Average width | Median width |
| 10 | 90.1 | 40.98 | 29.68 | 89.9 | 16.05 | 11.76 | 89.7 | 16.24 | 11.84 |
| 50 | 89.3 | 36.14 | 27.34 | 89.0 | 16.29 | 12.15 | 88.8 | 16.28 | 11.91 |
| 100 | 89.7 | 43.46 | 29.98 | 89.3 | 16.91 | 12.63 | 89.6 | 17.19 | 12.48 |
| 150 | 89.0 | 45.97 | 33.60 | 89.2 | 16.20 | 12.24 | 89.2 | 16.43 | 12.15 |
| 200 | 89.1 | 43.99 | 31.66 | 89.4 | 16.24 | 12.30 | 89.1 | 16.28 | 12.48 |

Table 8: The ablation experimental results of window length $w$ in the Google stock dataset with nominal level $\alpha = 10\%$.

| $w$ | Prophet Model | | | AR Model | | | Theta Model | | |
|---|---|---|---|---|---|---|---|---|---|
| | Coverage (%) | Average width | Median width | Coverage (%) | Average width | Median width | Coverage (%) | Average width | Median width |
| 10 | 89.9 | 50.33 | 42.47 | 91.7 | 22.74 | 22.05 | 89.6 | 33.49 | 29.42 |
| 50 | 89.4 | 47.38 | 40.90 | 90.2 | 20.69 | 19.21 | 89.3 | 31.63 | 29.64 |
| 100 | 89.8 | 53.12 | 44.36 | 89.7 | 19.84 | 17.63 | 89.6 | 30.71 | 28.11 |
| 150 | 89.1 | 55.62 | 47.59 | 90.0 | 20.53 | 17.85 | 89.7 | 30.54 | 28.98 |
| 200 | 89.4 | 53.65 | 45.49 | 89.8 | 20.48 | 18.02 | 89.8 | 30.99 | 30.06 |

$\mathcal{N}(0, I_4)$, $\epsilon_t \sim \mathcal{N}(0, 1)$. And we set:

$$\beta_t = \beta^{(0)} = (2, 1, 0, 0)^\top, \quad t = 1, \ldots, 500,$$
$$\beta_t = \beta^{(1)} = (0, -2, -1, 0)^\top, \; t = 501, \ldots, 1500,$$
$$\beta_t = \beta^{(2)} = (0, 0, 2, 1)^\top, \quad t = 1501, \ldots, 2000,$$

where two changes in the coefficients happen up to time 2000.

We compare ECI and its variants with some competing methods about the coverage and prediction set width. Figures 10 and 11 show the result of coverage and set width, while the base predictor is Theta model.

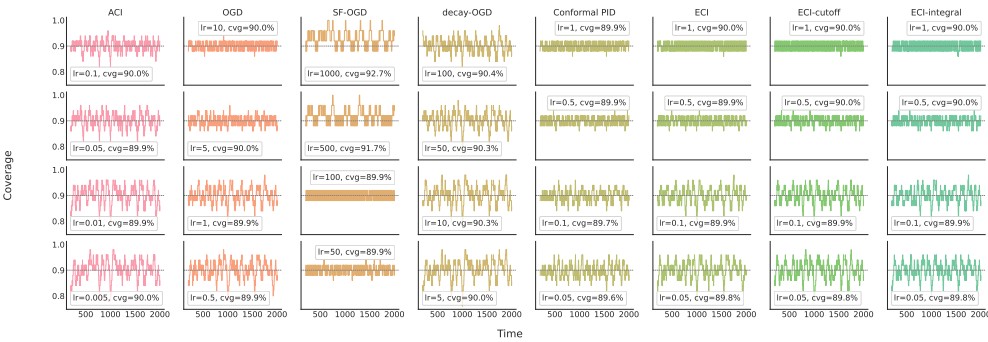

Figure 10: Coverage result on synthetic data under a changepoint setting.

## F  EXPERIMENTAL RESULTS WITH TRANSFORMER

As one of the most successful deep learning models, Transformer has had a significant impact on various application fields, including time series. Thus it is interesting to conduct an additional

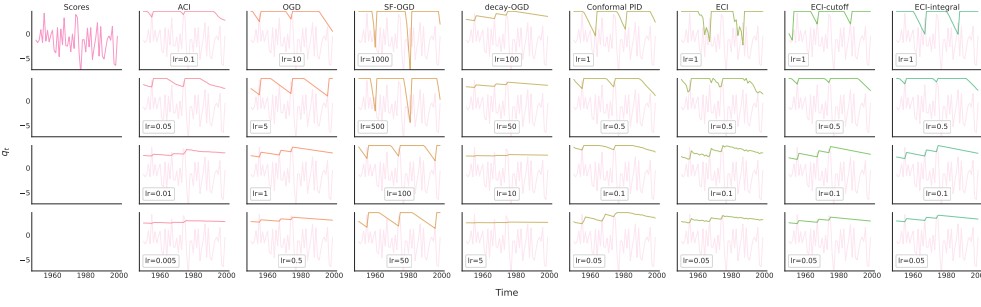

Figure 11: Set width result on synthetic data under a changepoint setting.

experiment with Transformer as the base model. We set input length to 12, output length to 1, the number of encoder layers to 3, the number of decoder layers to 3, and the number of features in the encoder/decoder inputs to 64.

The quantitative results are shown in Tables 9 and 10. Consistent with other base models, ECI variants have achieved the best performance on five benchmark datasets with Transformer.

Table 9: The experimental results with Transformer base model at nominal level $\alpha = 10\%$.

| Method | Amazon stock | | | Google stock | | | Electricity demand | | |
|---|---|---|---|---|---|---|---|---|---|
| | Coverage ( %) | Average width | Median width | Coverage ( %) | Average width | Median width | Coverage ( %) | Average width | Median width |
| ACI | 90.1 | $\infty$ | 40.44 | 90.2 | $\infty$ | 57.13 | 90.2 | $\infty$ | 0.109 |
| OGD | 89.4 | 52.68 | 31.00 | 90.1 | 109.27 | 89.00 | 90.1 | 0.139 | 0.120 |
| SF-OGD | 89.3 | 56.56 | 31.75 | 90.1 | 88.30 | 70.55 | 90.3 | 0.141 | 0.114 |
| decay-OGD | 89.8 | 93.16 | 34.98 | 89.9 | 120.25 | 69.81 | 90.3 | 0.147 | 0.111 |
| PID | 89.8 | 55.36 | 39.04 | 90.1 | 78.69 | 58.65 | 89.9 | 0.428 | 0.435 |
| ECI | 89.9 | 49.07 | 33.79 | 89.9 | 70.93 | 55.00 | 90.2 | 0.135 | 0.111 |
| ECI-cutoff | 89.7 | **45.01** | 29.64 | 89.9 | **66.67** | **51.29** | 89.9 | **0.133** | **0.108** |
| ECI-integral | 89.7 | 45.02 | **29.46** | 90.0 | 68.64 | 52.45 | 90.2 | 0.135 | 0.111 |

Table 10: The experimental results with Transformer base model at nominal level $\alpha = 10\%$.

| Method | Amazon stock | | | Google stock | | |
|---|---|---|---|---|---|---|
| | Coverage ( %) | Average width | Median width | Coverage ( %) | Average width | Median width |
| ACI | 90.3 | $\infty$ | 11.69 | 89.9 | $\infty$ | 8.20 |
| OGD | 89.9 | 9.72 | 9.50 | 89.9 | 8.13 | 8.20 |
| SF-OGD | 90.0 | 10.93 | 9.97 | 90.0 | 12.55 | 11.65 |
| decay-OGD | 89.7 | 14.43 | 11.47 | 90.1 | 8.30 | 8.24 |
| PID | 89.9 | 10.02 | 9.75 | 89.7 | 10.75 | 9.05 |
| ECI | 89.9 | 9.15 | 9.16 | 89.9 | 8.09 | 8.15 |
| ECI-cutoff | 90.0 | 10.42 | 9.95 | 89.9 | **8.01** | **8.10** |
| ECI-integral | 89.9 | **8.87** | **8.60** | 89.9 | 8.04 | 8.13 |

# G    MORE DETAILS ON EXPERIMENTS

## G.1    DISCUSSION ON THE SCORECASTER

It is worth noting that the Conformal PID baseline outperforms ECI under the Prophet base model in Table 3. Actually, the scorecaster term of conformal PID in the baseline utilizes the relatively

accurate Theta model (which can also be replaced by AR or Transformer) to take advantage of any leftover signal and residualize out systematic errors in the score distribution. It can be regarded as an additional component that "sits on top" of the base forecaster (base model). Therefore, across all test datasets, the performance of conformal PID in the Prophet model consistently outperforms that of the AR and Theta model. This is attributed to the fact that Prophet, being a worse-performing model, is complemented by the superior performance of PID's scorecaster Theta, which is a better-performing model.

We maintained the settings of Table 3 to test the performance of ECI combined with the scorecaster (Theta model), as shown in Table 11. The results demonstrated that ECI+scorecaster is consistently superior over Conformal PID across three base models, thereby validating the effectiveness of the ECI update. Interestingly, for the worse-performing Prophet base model, adding the scorecaster enhanced the performance of ECI, while the scorecaster tended to degrade performance when better-performing base models were used. This observation is also mentioned in Angelopoulos et al. (2023b): "an aggressive scorecaster with little or no signal can actually hurt by adding variance to the new score sequence".

Table 11: The experimental results in the electricity demand dataset with nominal level $\alpha = 10\%$. Both scorecasters of PID and ECI+scorecaster are Theta model.

| Method | Prophet Model | | | AR Model | | | Theta Model | | |
|---|---|---|---|---|---|---|---|---|---|
| | Coverage ( %) | Average width | Median width | Coverage ( %) | Average width | Median width | Coverage ( %) | Average width | Median width |
| PID | 90.1 | 0.207 | 0.177 | 90.0 | 0.434 | 0.432 | 89.9 | 0.413 | 0.411 |
| ECI | 90.0 | 0.384 | 0.382 | 90.0 | **0.117** | **0.098** | 89.9 | **0.071** | **0.055** |
| ECI+scorecaster | 90.3 | **0.193** | **0.166** | 90.0 | 0.420 | 0.428 | 89.9 | 0.395 | 0.409 |

### G.2 MORE DETAILS ON LEARNING RATES IN THE EXPERIMENTS

In this section, we show more comprehensive experimental results. We set base predictor as Theta model and all experimental setting are aligned with Section 4. Coverage and prediction set results can be seen as the following Figure 12 to Figure 19.

Since the methods based on OGD are highly sensitive to the learning rate, we initially select four appropriate learning rates for these methods across various datasets, and then choose the one that performs the best among these four. In fact, aside from the OGD-based methods, the four learning rates for other methods remain unchanged. We compile a list that includes all the learning rates, which are

$$\text{ACI} : \eta = \{0.1, 0, 05, 0.01, 0.005\},$$
$$\text{OGD} : \eta = \{10, 5, 1, 0.5, 0.1, 0.05, 0.01, 0.005\},$$
$$\text{SF-OGD} : \eta = \{1000, 500, 100, 50, 10, 5, 1, 0.5, 0.1, 0.05\},$$
$$\text{decay-OGD} : \eta = \{2000, 1000, 200, 100, 20, 10, 2, 1, 0.2, 0.1\},$$
$$\text{Conformal PID} : \eta = \{1, 0.5, 0.1, 0.05\},$$
$$\text{ECI} : \eta = \{1, 0.5, 0.1, 0.05\},$$
$$\text{ECI-cutoff} : \eta = \{1, 0.5, 0.1, 0.05\},$$
$$\text{ECI-integral} : \eta = \{1, 0.5, 0.1, 0.05\}.$$

Note that, except ACI and OGD, other methods use $\eta_t$ as adaptive learning rate in practice. Specifically, for SF-OGD:

$$\eta_t = \eta \cdot \frac{\nabla \ell^{(t)}(q_t)}{\sqrt{\sum_{i=1}^{t} \|\nabla \ell^{(i)}(q_i)\|_2^2}},$$

where $\ell^{(t)}(q_t)$ is quantile loss and $q_t$ is the predicted radius at time $t$. For decay-OGD:

$$\eta_t = \eta \cdot t^{-\frac{1}{2} - \epsilon},$$

where the hyperparameter $\epsilon = 0.1$ follows Angelopoulos et al. (2024). For conformal PID, ECI, ECI-cutoff and ECI-integral:

$$\eta_t = \eta \cdot (\max\{s_{t-w+1}, \cdots, s_t\} - \min\{s_{t-w+1}, \cdots, s_t\}),$$

where $s_t$ is the non-conformality score at time $t$ and the window length $w = 100$ follows Angelopoulos et al. (2023b).

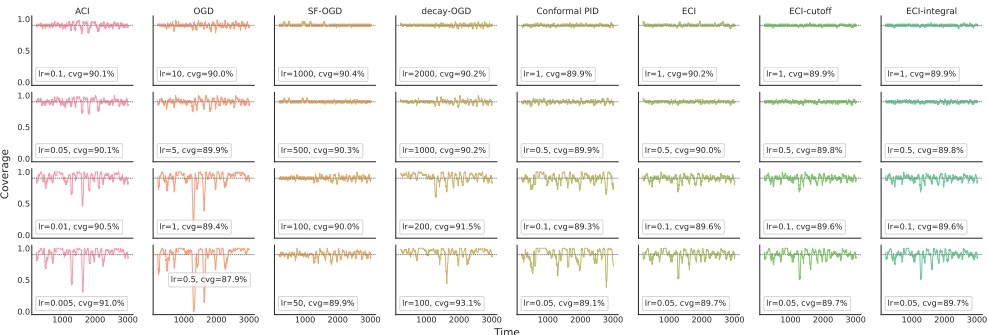

Figure 12: Coverage result on Google stock dataset.

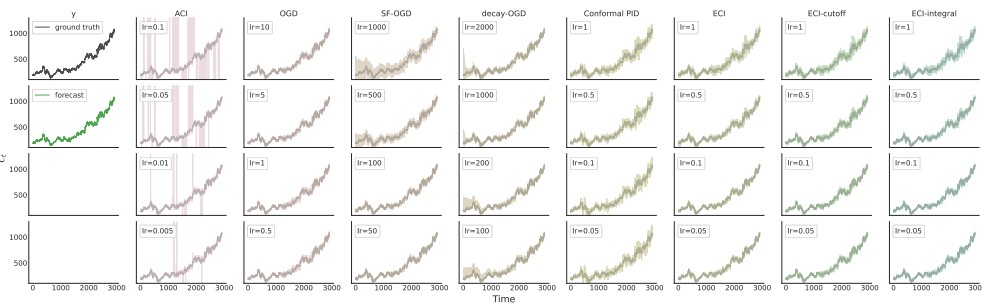

Figure 13: Prediction set result on Google stock dataset.

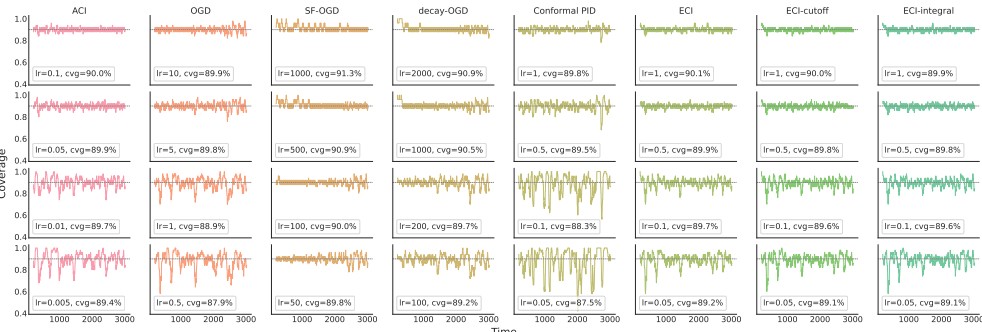

Figure 14: Coverage result on Amazon stock dataset.

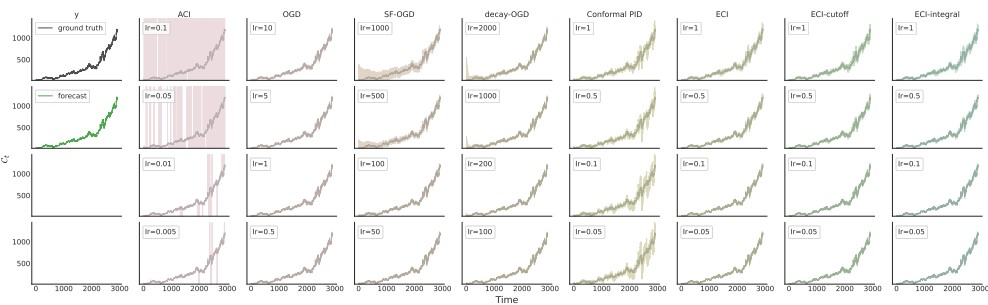

Figure 15: Prediction set result on Amazon stock dataset.

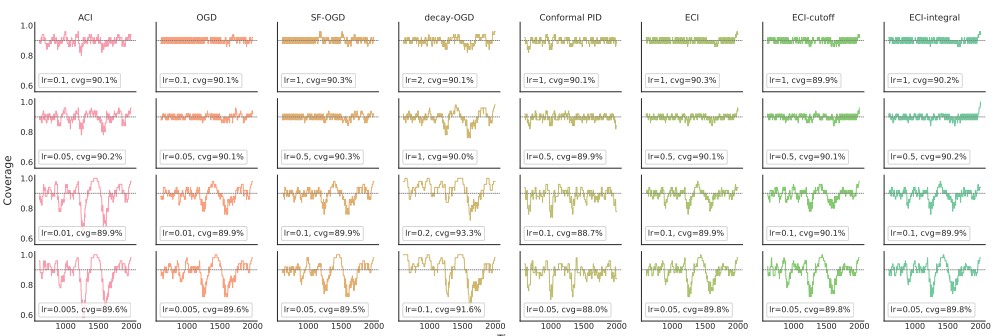

Figure 16: Coverage result on electricity demand dataset.

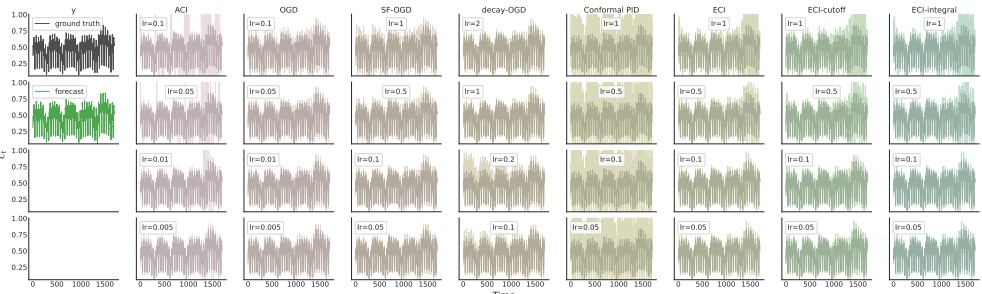

Figure 17: Prediction set result on electricity demand dataset.

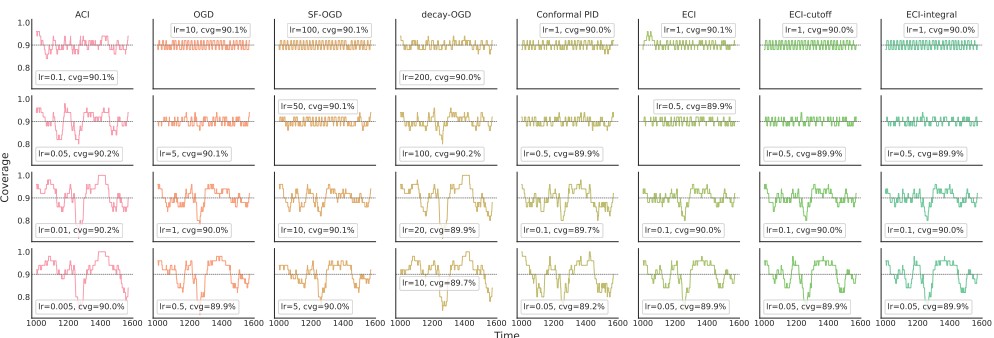

Figure 18: Coverage result on Delhi temperature dataset.

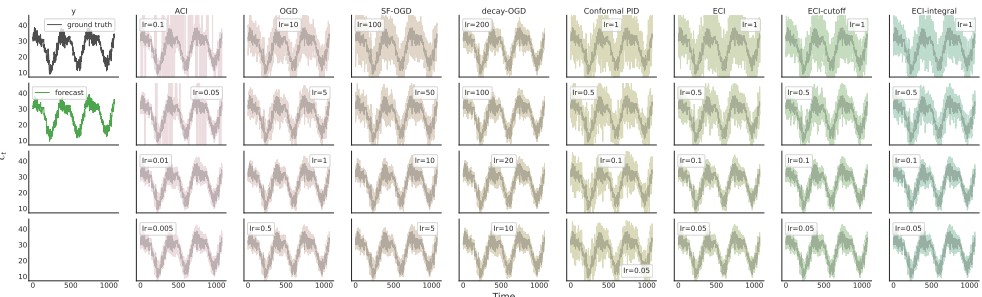

Figure 19: Prediction set result on Delhi temperature dataset.

