# OpenReview forum: "Error-quantified Conformal Inference for Time Series"
_ICLR.cc/2025/Conference — ICLR 2025 Poster_

### Official Review · Reviewer_Yq3J · 2024-11-03

**Soundness:** 3
**Presentation:** 3
**Contribution:** 3
**Rating:** 6
**Confidence:** 5

**Summary:**

The author propose an extension over the Adaptive Conformal Inference method of Gibbs and Candes, then used in the time-series realm by Zaffran et al. by incorporating in the adaptation procedure information about the magnitude of the error in the miscoverage rate.
After having stated the method and analysed its performance with some real-world applications.

**Strengths:**

The proposed extension is rather interesting, seems to provide reasonable results, and puts itself in a rather interesting area of conformal prediction.

**Weaknesses:**

- "Conformal Inference for Time Series" is an extremely broad title, which gives the impression that the authors are the first in doing something in this which is, in fact, not true.

- I found the article written in a very rough and unrefined way: I have counted at least 10 instances of missing articles (the...). Moreover, the introduction contains claims that are either too bold, or require a deeper explanation. What does it mean, for instance, that "Bayesian recurrent neural networks or deep gaussian processes are difficult to calibrate?", or, why is it meaningful to say that "quantile regression models may overfit when estimating uncertainty"

- I am seriously concerned about the "stock market" application. It seems to me that the authors aim at forecasting log stock prices, which are known to be nonstationary. In fact, the usual approach in the econometric literature is to forecast returns (i.e. log differences) which instead, tend to be a stationary time series.

**Questions:**

- The "adaptive" CI school is only one of the possible approache to deal with the absence of indipendence between observations in Conformal Inference, in fact, https://proceedings.mlr.press/v75/chernozhukov18a.html provide a fairly interesting approach, which would be interesting to see compared to the authors proposal.

- A very recent contribution by Oliveira et al. https://jmlr.org/papers/v25/23-1553.html shows that the contribution of adaptivity procedures to be negligible. How do the authors comment about this?

---

> ### Author Response · Authors · 2024-11-20
>
> >"Conformal Inference for Time Series" is an extremely broad title, which gives the impression that the authors are the first in doing something in this which is, in fact, not true.
>
> The latter half of our title describes the domain and the methodology, while the former half indicates our insight into the deficiency of existing methods in conformal inference for time series.
>
> >I found the article written in a very rough and unrefined way: I have counted at least 10 instances of missing articles (the...). Moreover, the introduction contains claims that are either too bold, or require a deeper explanation. What does it mean, for instance, that "Bayesian recurrent neural networks or deep gaussian processes are difficult to calibrate?", or, why is it meaningful to say that "quantile regression models may overfit when estimating uncertainty"
>
> Thank you for the reminder. We have checked the written and corrected errors in our manuscript.
>
> "Bayesian recurrent neural networks or deep gaussian processes are difficult to calibrate?": The reason why they are hard to calibrate is that they can not provide corresponding prediction sets based on some specified confidence level as conformal prediction does. In contrast, conformal predictions have become popular because they are model-agnostic and able to enhance complex pre-trained models with predictive uncertainties post hoc. Hence, conformal methods can yield well-calibrated prediction sets based on the output of any point estimation model, irrespective of the base model’s structure.
>
> "Quantile regression models may overfit when estimating uncertainty": They may capture the noise in the training data rather than the true data distribution, especially when the amount of data is limited and  the model is inaccurately specified. In addition, the non-smoothness of quantile regression and its sensitivity to minor changes in the data can also lead to overfitting.
>
>
> >I am seriously concerned about the "stock market" application. It seems to me that the authors aim at forecasting log stock prices, which are known to be nonstationary. In fact, the usual approach in the econometric literature is to forecast returns (i.e. log differences) which instead, tend to be a stationary time series.
>
> For fair comparison, we follow the stock experimental setup of conformal PID. Besides, our methods have no requirement for data distribution, including stationarity. For practical use, we can deploy a model that offers more accurate forecasts of returns. However, conformal methods are model-free, which means that we quantify uncertainties on top of the forecasting models, regardless of the quality of them.
>
>
> >The "adaptive" CI school is only one of the possible approache to deal with the absence of independence between observations in Conformal Inference, in fact, https://proceedings.mlr.press/v75/chernozhukov18a.html provide a fairly interesting approach, which would be interesting to see compared to the authors proposal.
>
> This paper extended conformal inference to dependent data by proposing a randomization procedure via permutation on blocking structures. However,  this method needs transformations of data to be a stationary and strong mixing series (see conditions (A) and (E) in Section 3.2 and Lemma 1 in Section 3.3 therein). Moreover, implementing the blocking and permutation method entails heavy computation, which is not appropriate for online settings.
>
> >A very recent contribution by Oliveira et al. https://jmlr.org/papers/v25/23-1553.html shows that the contribution of adaptivity procedures to be negligible. How do the authors comment about this?
>
> Oliveira et al. (2024) studied the split conformal prediction method for non-exchangeable data. However, the theoretical coverage results rely on the assumptions on the distribution of data, such as $\beta$-mixing condition, which is very hard to verify in complex time series data. Besides, we politely disagree that Oliveira et al. (2024) shows the contribution of adaptivity procedures is negligible. First, Oliveira et al. (2024) only conducted the numerical comparison with one adaptive procedure (DtACI) in one real data. Second, even under the same assumptions, Oliveira et al. (2024) did not show that split conformal prediction can strictly perform better than adaptive procedures in theory.

---

> > ### Comment · Reviewer_Yq3J · 2024-11-22
> >
> > In fairness, I am only partially satisfied with the answers.
> > I do not find the authors explanation on their choice of a title convincing. Like this, a reader unaware that there is a rich body of literature (following, by the way, several different approaches) would still believe that the authors invented the field of Conformal Time Series forecasting themselves.
> > Something along the line of "Error-Quantified Conformal PID Forecasting" would be more honest, and representative of the actual work.
> > Bayesian recurrent neural networks and deep gaussian processes can be calibrated using conformal machinery as easily as Prophet, or the other models used by the authors. That's why the statement left me a bit baffled. I do not think it is a central part of the argument... just a wrong statement.
> > The same applies for what is said about quantile regression. "Overfitting" seems to refer to the use of interpolating models, which seems out of place here. If the authors refer to undercoverage and overcoverage when talking about probabilistic calibration, I believe it is better to change the wording. Anyways, empirical studies (e.g. https://proceedings.neurips.cc/paper_files/paper/2019/file/5103c3584b063c431bd1268e9b5e76fb-Paper.pdf) show that the situation is simply not clear. Again, not a fundamental part of the argument, just a wrong statement.
> >
> > With respect to the second question, while it is true that Conformal methods are model free, you usually have an assumption of iid-ness. The identicality of the distribution is simply broken in the non-stationary case. Being a very non-standard situation for time-series forecasting, I would appreciate a bit more clarity by the authors, and a discussion on why predicting a nonstationary time series is possible via their method.
> >
> > I have appreciated the reply on the Chernozhukov approach... but I wonder why the authors have not inserted it in the paper! Chernozhukov et al 2018 is in fact the first paper dealing with CP for dependent data, so a comment on it would help in putting the authors contribution in the right scientific context.
> >
> > With respect to Olivera, I beg to differ with the authors. Their results hold under very weak conditions on the closeness of empirical and population cdfs over calibration data, and between training and test data.
> > Then, they show as a general case that beta-mixing conditions imply the hypotheses that the authors use to prove their results.
> > Again, this does not mean that the authors' work is meaningless, but I would appreciate the authors mentioning this, and including this in the paper.

---

> > > ### Author Response · Authors · 2024-11-23
> > >
> > > >Choice of the title and statements
> > >
> > > Thank you for the suggestions. We have modified the title to “Error-quantified Conformal Inference for Time Series” in the new pdf. We also modified the statements to avoid confusion.
> > >
> > > >Discussion on why predicting a nonstationary time series is possible via their method.
> > >
> > > Requiring no assumption on the distribution of data is the most impressive point of the adaptive procedure.
> > >
> > > In scenarios where exchangeability (or i.i.d) assumption does not hold, such as time series, it is very difficult to achieve the real-time coverage guarantee $P(Y\_{t+1} \in \hat{C}\_{t+1}(X\_{t+1})) \geq 1 \- \alpha$ without any other constraints.
> > >
> > > Alternatively, works based on ACI consider the following long-term miscoverage control:
> > > $$\frac1T\sum\_{t=1}^T 1(Y\_t\notin\hat{C}\_t(X\_t))=\alpha.$$
> > >
> > > With the adaptive procedure, achieving the long-term miscoverage control does not need any distributional assumption of data. We can refer to the proof of Proposition 4.1 in [1] and Proposition 1 in [2]. Also, the proof of Theorem 1 and 2 in our paper does not need distributional assumption of data.
> > >
> > > [1] Gibbs I, Candes E. Adaptive conformal inference under distribution shift[J]. Advances in Neural Information Processing Systems, 2021, 34: 1660-1672.
> > >
> > > [2] Angelopoulos A, Candes E, Tibshirani R J. Conformal pid control for time series prediction[J]. Advances in neural information processing systems, 2024, 36.
> > >
> > >
> > > >Two related papers
> > >
> > > We are willing to do it. We have mentioned the two papers and given brief comments on them (see line 121-123). We hope our responses answer the questions sufficiently to earn your support. Please let us know how we can improve if you still feel we are missing something.

---

> > > > ### Comment · Reviewer_Yq3J · 2024-11-25
> > > >
> > > > Thank you for the kind reply.
> > > > Now the paper and the contribution is much clearer.
> > > > I still find the work relatively derivative, so I am going for a 6. Good luck with the selection!

---

### Official Review · Reviewer_FWjo · 2024-11-04

**Soundness:** 3
**Presentation:** 3
**Contribution:** 2
**Rating:** 6
**Confidence:** 3

**Summary:**

This work introduces Error-quantified Conformal Inference (ECI), a scheme for providing prediction intervals for time series data. Standard conformal inference requires exchangeability for statistical coverage guarantees to hold but this is generally violated in the context of time series data. Other methods to provide prediction intervals exist but many do not incorporate the magnitude of miscoverage when providing updates for future intervals. ECI utilizes this information for its updates. The authors derive a coverage guarantee for their methodology and  provide empirical results on synthetic and real-world datasets to support their claims. Compared to baselines that do not incorporate the magnitude of error they find that they are able to provide narrower intervals.

**Strengths:**

- General presentation of the problem is clear and this is an approach could be of interest to an active community
- Mathematical derivations are included
- Strong empirical results on synthetic and real-world datasets

**Weaknesses:**

- Derivation on line 266 is difficult to read because it is inline
- Minor grammatical errors and occasional typos scattered throughout the manuscript
- See questions

**Questions:**

- Do the proofs for the validity of EQ-integral and EQ-cutoff need to be adjusted from the proof of EQI?
- For Assumption 1 should it also be "$\forall t$"
- Line 159: "$(1-\alpha) (1 + t^{-1})$" is not necessarily in $[0, 1]$. For instance if $t=3$ and $\alpha = 0.001$. How can this be a sample quantile?
- The EQI intervals visually are much more jagged. Could this ever result in dangerous overfitting?
- The significance of Figure 2 is unclear to me and the caption does not provide much detail. Why is this important and what is going on in this figure?
- Would error magnitude information be more useful for better or worse performing base models $f$ relative to other methods (ACI, OGD, etc)?

---

> ### Author Response · Authors · 2024-11-20
>
> Thank you for the thoughtful review, and we enjoyed your helpful comments!
>
> Please see the main response (in a separate comment) for a summary of the extensive improvements we have made to the paper. We hope our responses answer the questions sufficiently to earn your support. Please let us know how we can improve if you still feel we are missing something.
>
> >Derivation on line 266 is difficult to read because it is inline. Minor grammatical errors and occasional typos scattered throughout the manuscript. For Assumption 1 should it also be "∀t".
>
> Thank you for the careful reading!  We have separated the inequality on line 266 into a single column for easier reading. We also checked the written and corrected errors.
>
> >Do the proofs for the validity of EQ-integral and EQ-cutoff need to be adjusted from the proof of EQI?
>
> Our proof of both Theorem 1 and Theorem 2 primarily lie in quantifying the EQ term $(s_t-q_t) \nabla f(s_t-q_t)$. Considering ECI-cutoff, all inequalities in the proof still hold after we replace it with the cutoff EQ term $(s_t-q_t) \nabla f(s_t-q_t)$. The assumptions made in Theorem 1 are also easier to satisfy.
>
> Hence the proof for ECI-cutoff does not need any adjustment. When it turns to the ECI-integral cases, ways of measuring inequalities in the original proof must be adjusted according to specific $w_i$. The proof will be more cumbersome. Hence, similar to ACI and Conformal PID, we only prove the ECI form that implies the main mechanism of our method.
>
> >Line 159: "(1−α)(1+1/t)" is not necessarily in [0,1]. For instance if t=3 and α=0.001. How can this be a sample quantile?
>
> Since $t$ represents the size of observed data, which is generally taken to be large compared to $1/\alpha$, $(1-\alpha)(1+1/t)$ will not exceed 1. If $(1-\alpha)(1+1/t)>1$,  the procedure will set the sample quantile to be 1.
>
> >The EQI intervals visually are much more jagged. Could this ever result in dangerous overfitting?
>
> We understand your concern. Actually, “jags” come from quick reactions to past mistakes. This is the common property of works based on adaptive procedure in conformal inference, and similar phenomenons can be found in ACI, conformal PID, etc (see Figure 4). If our prediction set correctly covers $Y_t$, we narrow the next prediction interval by a small size (i.e. $\alpha-EQ$). On the contrary, once there is a miscoverage point, we widen the next prediction set by a relatively large value (i.e. $1-\alpha + EQ$) and make the several following points to be covered. Therefore, whenever we widen our interval (due to miscoverage) and then narrow it in the next steps (due to coverages), there is a jag.
>
> Besides, the reason why SF-OGD and decay-OGD seem visually to be more “smooth” is due to the fact that their learning rate is decreasing and close to zero after several iterations, causing the length of predictive intervals to be constant (we can see the jags in the early stages of SF-OGD and decay-OGD in Figure 4). However, when the learning rate is close to zero, the methods cannot react to  sudden distribution shifts.
>
>
> > The significance of Figure 2 is unclear to me and the caption does not provide much detail. Why is this important and what is going on in this figure?
>
>
> We aim to visualize the EQ term and provide readers with a clearer understanding of the role it plays and the shape it should take. Please see main response (2).
>
> >Would error magnitude information be more useful for better or worse performing base models f relative to other methods (ACI, OGD, etc)?
>
> The error magnitude is quantified by $s_t-q_t$,  which aims to minimize the interval size while ensuring coverage. Taking the symmetric case as an example, the ECI outputs the interval $[f(X_t)-c_t, f(X_t)+c_t]$.  A more effective algorithm would bring $f(X_t)$ close to $Y_t$, while ECI reduces $c_t$, the radius of interval.
>
> Our method imposes no specific requirements on the base model; for instance, the Prophet model may exhibit worse MSE, whereas the Theta model performs better. Importantly, we do not have prior knowledge regarding the effectiveness of the models, and our approach applies to both better-performing and worse-performing base models.

---

> > ### Author Response · Authors · 2024-11-25
> > **Looking forward to your feedback**
> >
> > Dear Reviewer,
> >
> > Thank you for reviewing our paper again! We have posted our responses to your questions and concerns. We are wondering if you have a new evaluation of our paper after reading our responses. If you have other questions, we are very happy to discuss them with you.
> >
> > Best,
> >
> > Authors

---

> ### Author Response · Authors · 2024-12-02
> **Discussion Reminder**
>
> Dear Reviewer FWjo,
>
> Thank you again for your time and effort in reviewing our paper. Since today is the final day for reviewers to post comments, we would appreciate your feedback on our responses. If you have any further questions, please let us know and we will be happy to provide additional clarifications. If not, we would be grateful if you could consider increasing your score to reflect the fact that we have addressed your concerns.
>
> Best regards,
>
> The Authors

---

### Official Review · Reviewer_f97D · 2024-11-04

**Soundness:** 3
**Presentation:** 3
**Contribution:** 2
**Rating:** 5
**Confidence:** 4

**Summary:**

This paper proposes Error-quantified Conformal Inference (ECI), based on the ACI approach, for online conformal inference with time series data. The key idea is to augment the online gradient descent with a smoothing term that reflects the extent of the gap between the non-conformity score and the threshold at each time step, moving beyond simple binary feedback. The main results include demonstrating coverage guarantees under a fixed learning rate, given bounded conditions for the smoothing function and non-conformity scores, and presenting a reasonable bound on the coverage gap in the case of well-designed adaptive learning rates.

**Strengths:**

- While the methods are relatively straightforward, they are explained clearly and precisely.
- The use of smoothing feedback to actively incorporate the degree of miscoverage during the update is a compelling idea, allowing the prediction sets to adapt well to, e.g., potential distributional shifts in time series data.
- Theoretical results are based on reasonable assumptions.

**Weaknesses:**

- The authors primarily select the Sigmoid function for smoothing. However, as illustrated in Figure 2, when the extent of miscoverage exceeds a certain level, the EQ term actually decreases (which is also the case for the Gaussian error function). While the authors justify this with the goal of ensuring robustness, the rationale seems somewhat unconvincing.
- In connection with the above, if the Sigmoid function is used, the parameter $c$ can be considered a hyperparameter. Also given that Theorem 1 imposes constraints on $c$ for coverage guarantees, a more in-depth discussion on its selection, beyond the brief mention in Appendix D, would be appreciated.
- The interpretation of the miscoverage bound presented in Theorem 2 is rather ambiguous. The choice of the learning rate is a crucial aspect, yet theoretical discussion on this remains superficial.
- Similarly, the experiments lack detailed discussion on how the learning rate, which is critical for the performance of each method, was chosen. The statement on Line 330, “we select the most appropriate range of learning rates $\eta$ for the respective datasets and present the best results in the tables,” is not only vague but could potentially raise concerns about data leakage. More transparency in the tuning process is needed.
- Additionally, the choices of $h$ and $w$ in the ECI-cutoff method are not discussed.

Overall, while the proposed method is straightforward and built on a solid idea, there is room for more detailed discussion regarding the rationales, interpretations of the results, and specific hyperparameter tuning.

Minor:
- Line 119: Please use \citep for Barber et al. (2021).
- Line 238: incorrect interval order
- Line 653: non-differential -> non-differentiable
- Are "coverage"s in, e.g., Figure 3, rolling averages?

**Questions:**

See weaknesses.

---

> ### Author Response · Authors · 2024-11-20
>
> >Overall, while the proposed method is straightforward and built on a solid idea, there is room for more detailed discussion regarding the rationales, interpretations of the results, and specific hyperparameter tuning.
>
> Thank you for the summary, and we appreciate your suggestions! Please see the main response (in a separate comment) for a summary of the extensive improvements we have made to the paper. We hope our responses answer the questions sufficiently to earn your support. Please let us know how we can improve if you still feel we are missing something.
>
> >The authors primarily select the Sigmoid function for smoothing. However, as illustrated in Figure 2, when the extent of miscoverage exceeds a certain level, the EQ term actually decreases (which is also the case for the Gaussian error function). While the authors justify this with the goal of ensuring robustness, the rationale seems somewhat unconvincing.
>
> Thanks for your feedback! Let us clarify our point further. Please see our main response (2).
>
> >In connection with the above, if the Sigmoid function is used, the parameter c can be considered a hyperparameter. Also given that Theorem 1 imposes constraints on c for coverage guarantees, a more in-depth discussion on its selection, beyond the brief mention in Appendix D, would be appreciated.
>
> We understand your concern. The constraints we impose on $c$ in Theorem 1 are for the convenience of the proof. Moreover, our experimental results in new Appendix D show that as $c$ varies, the changes in both coverage and width are relatively small and the two metrics are actually a trade-off. It is worth noting that, for fixed coverage, the width of our methods with $c \in $ {0.1, 0.5, 1, 1.5, 2} are **consistently shorter** than other methods (see Table 1, Figure 8 and 9). We choose $c=1$ that has good performance in all our datasets. A more effective value of $c$ can be obtained by grid search.
>
> >The interpretation of the miscoverage bound presented in Theorem 2 is rather ambiguous. The choice of the learning rate is a crucial aspect, yet theoretical discussion on this remains superficial.
>
> Theorem 2 provides the coverage results of ECI under arbitrary learning rates (the worst-case scenario), but **it illustrates how different learning rates influence the bounds**, which cannot be directly used to choose the learning rates. In practice, we have adopted the same automatic choice of the learning rate $\eta_t$ as conformal PID which has good performance across our diverse datasets. We advocate for its application in practical scenarios, thereby eliminating the need for manual tuning.
>
> Another approach to choosing learning rates is to utilize methods from online learning literature—run multiple versions of the adaptive procedure with different choices of learning rates $\eta$ and take weighted sums to get the final result. We list some related works below. These techniques can also be used in our proposed method.
>
> [1] Gibbs I, Candès E J. Conformal inference for online prediction with arbitrary distribution shifts[J]. Journal of Machine Learning Research, 2024, 25(162): 1-36.
>
> [2] Bhatnagar A, Wang H, Xiong C, et al. Improved online conformal prediction via strongly adaptive online learning[C]//International Conference on Machine Learning. PMLR, 2023: 2337-2363.
>
> [3] Hajihashemi E, Shen Y. Multi-model Ensemble Conformal Prediction in Dynamic Environments[C]//The Thirty-eighth Annual Conference on Neural Information Processing Systems, 2024.

---

> > ### Author Response · Authors · 2024-11-20
> >
> > >Similarly, the experiments lack detailed discussion on how the learning rate, which is critical for the performance of each method, was chosen. The statement on Line 330, “we select the most appropriate range of learning rates η for the respective datasets and present the best results in the tables,” is not only vague but could potentially raise concerns about data leakage. More transparency in the tuning process is needed.
> >
> > We are sorry for the confusion. Since the methods based on OGD are highly sensitive to the learning rate, we select 8 learning rates for OGD and 10 learning rates for SF-OGD and decay-OGD across various datasets. Then we choose the one that performs the best (shortest average width with coverage > 89%). And we select 4 learning rates for ACI, conformal PID, and our methods. We compile a list that includes all the learning rates for each method in Appendix G.2.
> >
> > The choice of the learning rate is a crucial task and we actually need some priors to guide our choices. Here are some insights: **higher learning rates** can help to control coverage more intensely, but meanwhile, may potentially lead to overreaction and a wider interval; whereas **lower learning rates** might reduce coverage due to an inability to respond to distribution shifts.
> >
> > Take ECI in Amazon and Google stock dataset for example (see Figure 12-15), for Prophet model (worse-performing), the best choices of $\eta$ are both 0.5; and for Theta model (better-performing), both 0.1. As for the synthetic dataset, which is easy to predict, the best choices of $\eta$ are 0.05 for all base models. Therefore, in practical use, we can pre-evaluate the model's accuracy in predicting; if the target data could be predicted accurately, we choose a smaller learning rate.
> >
> >
> > >Additionally, the choices of h and w in the ECI-cutoff method are not discussed.
> >
> > We are willing to discuss them. Firstly, setting the cutoff justifies the assumption made in Theorem 1 that requires the upper bound of $\nabla f$ to be sufficiently small (see lines 267-271). Secondly, to avoid manual tuning, we recommend $h_t$ that performs well across all our datasets:
> >
> > $$h_t=h \max_{t-w\leq i,j \leq t}|s_i-s_j|,$$
> >
> > where we take $h=1$ by default.
> >
> > Due to the high diversity of the datasets, there is no universal choice for $w$. In our paper, to ensure experimental fairness, we set $w=100$ in alignment with the conformal PID. For a specific dataset, the optimal $w$ can be obtained through grid search.
> >
> > Also, we are interested in studying the performance of different $w$ across various datasets. We have shown the results of ECI-cutoff with different choices of $w$ in Amazon, Google and the synthetic dataset in Appendix D.2. In general, $w=100$ has the shortest average width with coverage greater than 89.5%. It is worth noting that in the synthetic dataset (see Table 6), there is a clear trend of increasing width and coverage as $w$ increases. This is because the main influencing factor at this time is the learning rate, and an increase in $w$ leads to a larger adaptive learning rate.
> >
> >  >Minor
> >
> > Thank you for the helpful fixes! The "coverage"s in figures and tables are computed by rolling average with window=50, aligned with previous works. We have already added the explanation in Figure 1.

---

> ### Author Response · Authors · 2024-11-25
> **Looking forward to your feedback**
>
> Dear Reviewer,
>
> Thank you for reviewing our paper again! We have posted our responses to your questions and concerns. We are wondering if you have a new evaluation of our paper after reading our responses. If you have other questions, we are very happy to discuss them with you.
>
> Best,
>
> Authors

---

> > ### Comment · Reviewer_f97D · 2024-11-25
> >
> > Thank you for the detailed response. Many of my concerns have been addressed. However, one remaining issue is the choice of the initial learning rate, which is a hyperparameter and appears to significantly impact the performance of ECI according to Figures 12–19. My understanding is that the best result among the choices of hyperparameters was reported, which does not fully alleviate my concerns regarding data leakage. Please correct me if I am wrong. If this issue is resolved, I would be inclined to increase my score.

---

> > > ### Author Response · Authors · 2024-11-30
> > > **Looking forward to your reply**
> > >
> > > Dear Reviewer,
> > >
> > > Thank you for the kind reply! We have posted our responses to your last issue. We are wondering if you have a new evaluation of our paper after reading our responses. We hope our responses answer the questions sufficiently to earn your support. If you have other questions, we are very happy to discuss them with you.
> > >
> > > Best,
> > >
> > > Authors

---

> ### Author Response · Authors · 2024-11-26
>
> Thank you for the kind reply. We agree that the selection of the initial learning rate $\eta$ is very important. This indeed is another branch for improving the adaptive procedure. (e.g. re-weighting approaches in  \[1,2,3\]).
>
>
> Honestly, the reason why we report the best results is to ensure fairness and validity of our experiments, since OGD-based methods are highly sensitive to the learning rate and can easily fail (see OGD with Prophet in table 2 below). To address your concern, we fixed the initial learning rate $\eta$ and conducted our experiments below. Specifically, ACI: $\eta=0.01$, OGD: $\eta=1$, SF-OGD: $\eta=100$, decay-OGD: $\eta=200$, conformal PID, ECI, ECI-cutoff, and ECI-integral: $\eta=0.1$.
>
> As our experiments show, even with a simple choice of $\eta \= 0.1$, ECI and its variants also outperform other methods in general. In the table below, the italic values represent coverage < 89%, and the bold values represent the shortest width with coverage > 89%.
>
> | Google |  | Prophet |  |  | AR |  |  | Theta |  |
> | :---: | :---: | :---: | :---: | :---: | :---: | :---: | :---: | :---: | :---: |
> | Method | Coverage | Average width | Median width | Coverage | Average width | Median width | Coverage | Average width | Median width |
> | ACI | 90.0 | $\infty$ | 66.83  | 89.8 | $\infty$ | 18.64  | 90.5 | $\infty$ | 32.78  |
> | OGD | *85.9* |  87.11  | 82.50 | 89.9 | **20.32** | 19.90  | 89.4 | 40.61 | 36.75 |
> | SF-OGD | 89.6 | 58.92  | 47.78  | 89.9 | 28.31  | 24.42  | 90.0 | 34.04  | 31.48  |
> | decay-OGD | 89.9 | 113.81 | 111.23 | 91.3 | 58.04 | 26.40 | 91.5 | 74.70  | 54.11 |
> | PID | 90.1 | 57.47  | 48.44  | 89.0 | 77.51 | 70.65 | 89.3 | 75.68 | 66.93 |
> | ECI | 89.3 | 57.46 | 49.31 | 89.8 | 22.39 | 18.59 | 89.6 | 30.92 | 29.53 |
> | ECI-cutoff | 89.0 | **55.69** | **48.29** | 89.7 | 21.70  | 18.20  | 89.6 | 30.71  | 28.11  |
> | ECI-integral | 89.0 | 56.69 | 48.50 | 89.7 | 21.59 | **18.10** | 89.6 | **30.42** | **28.02** |
>
>
> | Amazon |  | Prophet |  |  | AR |  |  | Theta |  |
> | :---: | :---: | :---: | :---: | :---: | :---: | :---: | :---: | :---: | :---: |
> | Method | Coverage | Average width | Median width | Coverage | Average width | Median width | Coverage | Average width | Median width |
> | ACI | 90.2 | $\infty$ | 46.97  | 89.8 | $\infty$ | 13.77  | 89.7 | $\infty$ | 12.31  |
> | OGD | *82.4* | 72.80 | 46.90 | 89.7 | 17.41 | 14.50 | *88.9* | 15.21 | 12.30 |
> | SF-OGD | 89.5 | 61.47  | **31.75**  | 89.9 | 24.44  | 21.05  | 90.0  | 23.88  | 21.14  |
> | decay-OGD | *85.7* | 80.5 | 48.59 | 89.7 | 20.23 | 14.01 | 89.7 | 21.02 | 14.36 |
> | PID | 89.1 | **60.28** | 48.34 | *88.3* | 91.30 | 51.00 | 88.3 | 91.73 | 51.54 |
> | ECI | *88.8* | 49.86 | 35.12 | 89.5 | 17.12  | 12.73  | 89.7 | 17.46 | 12.49 |
> | ECI-cutoff | *88.5* | 48.31 | 34.39 | 89.3 | **16.91**  | 12.63  | 89.6 | **17.19**  | 12.48  |
> | ECI-integral | *88.8* | 49.35 | 35.37 | 89.5 | 16.99  | **12.62**  | 89.6 | 17.20  | **12.46**  |
>
> In practice, we can select the learning rate based on prior knowledge, cross-validation or re-weighting approaches (as discussed in our previous comments). Take the re-weighting approach in \[1\] as an example, they deploy a candidate set of learning rates, run multiple versions of the adaptive procedure in parallel (each version corresponds to a learning rate), and construct prediction intervals based on historical performance of each.
>
> If you still have concerns, please let us know so that we can have a further discussion.
>
> \[1\] Gibbs I, Candès E J. Conformal inference for online prediction with arbitrary distribution shifts\[J\]. Journal of Machine Learning Research, 2024, 25(162): 1-36.
>
> \[2\] Bhatnagar A, Wang H, Xiong C, et al. Improved online conformal prediction via strongly adaptive online learning\[C\]//International Conference on Machine Learning. PMLR, 2023: 2337-2363.
>
> \[3\] Hajihashemi E, Shen Y. Multi-model Ensemble Conformal Prediction in Dynamic Environments\[C\]//The Thirty-eighth Annual Conference on Neural Information Processing Systems, 2024\.

---

> ### Author Response · Authors · 2024-12-02
> **Discussion Reminder**
>
> Dear Reviewer f97D,
>
> Thank you again for your time and effort in reviewing our paper. Since today is the final day for reviewers to post comments, we would appreciate your feedback on our responses. If you have any further questions, please let us know and we will be happy to provide additional clarifications. If not, we would be grateful if you could consider increasing your score to reflect the fact that we have addressed your concerns.
>
> Best regards,
>
> The Authors

---

### Official Review · Reviewer_P71t · 2024-11-07

**Soundness:** 3
**Presentation:** 3
**Contribution:** 3
**Rating:** 8
**Confidence:** 3

**Summary:**

This paper focuses on uncertainty quantification for time-series prediction, where exchangeability does not hold due to distribution shift and sequential dependence of data-points. Online conformal inference methods can be utilized for this task. Recent body of work on online conformal inference methods uses online gradient descent on the quantile loss function to update the thresholds of prediction sets. However, the subgradients of quantile loss function, that update the threshold at every time step, only leverage information on whether the outcome at that time-step is contained in the prediction set. The update does not include information on the magnitude of miscoverage. This paper introduces a smoothed version of quantile loss function such that the subgradient carries this information. Thus, 'Error-quantified Conformal Inference' incorporates a smooth version of feedback to update the threshold that defines prediction sets. The paper also introduces variants of smoothed ECI feedback for uncertainty quantification in time-series. Theoretical results on coverage guarantees and averaged miscoverage error are obtained without any assumptions on the data-generating distribution. Experimental results demonstrate superior performance of the ECI method in terms of reduced width of prediction sets while maintaining good coverage in comparison with other important online conformal inference baselines.

**Strengths:**

The idea of a smoothed version of quantile loss function to generate a more informative feedback for online gradient descent is simple and intuitive. It contributes meaningfully to improve uncertainty quantification in the online setting.

The mathematical intuition for the idea is well-presented and contextualized within the literature on online conformal inference.

The proposed technique produces consistent improvement in the width of prediction sets while maintaining coverage guarantees in a number of time series prediction tasks.

**Weaknesses:**

The authors mention difficulties in calibrating the outcomes of complex machine learning models like transformer, along with other models like deep Gaussian processes. However, the proposed techniques were evaluated on simpler time series models. It would be helpful to have some results with more complex models or discuss how the method would impact them.

It would be helpful to discuss the tradeoffs between different ECI variants in the context of experimental results.

For the place where the conformal PID baseline beats the proposed method, it would be helpful to explain the scorecaster term in a couple of lines so that it is more obvious why this baseline could produce a superior performance.

**Questions:**

For Table 3: Did the authors try to use a similar scorecaster term within the ECI update and check if that could improve over conformal PID in the Prophet model?

Also, is there any intuition for why decay-OGD baseline beats ECI in Table 5 (Prophet model)?

---

> ### Author Response · Authors · 2024-11-20
>
> >The authors mention difficulties in calibrating the outcomes of complex machine learning models like transformer, along with other models like deep Gaussian processes. However, the proposed techniques were evaluated on simpler time series models. It would be helpful to have some results with more complex models or discuss how the method would impact them.
>
> We are very willing to test the effectiveness of the more complex models like Transformer! The results can be seen in additional Appendix F (see Tables 9 and 10). Consistent with other base models, ECI variants have achieved the best performance on five benchmark datasets with Transformer as expected.
>
> Actually, conformal predictions have become popular because they are model-agnostic and able to apply to any underlying model and enhance it with predictive uncertainties post hoc. Hence, conformal methods can yield well-calibrated prediction sets based on the output of any point estimation model, irrespective of the base model’s structure.
>
> >It would be helpful to discuss the tradeoffs between different ECI variants in the context of experimental results.
>
> We fully agree! The ECI-integral can leverage all available past data, thus incorporating a greater amount of information, particularly in stock datasets (see Table 1 and 2). However, in the climate domain where the data is characterized by a shorter temporal horizon (see Table 4), the performance of ECI-integral may be less effective. As for ECI-cutoff, we believe it outperforms ECI-integral when there exists numerous outliers, causing $s_t-q_t$ to be exceedingly large (see Table 4).
>
> >For the place where the conformal PID baseline beats the proposed method, it would be helpful to explain the scorecaster term in a couple of lines so that it is more obvious why this baseline could produce a superior performance.
>
> Thank you for your advice! We have added some details in the experiment part.
>
> The scorecaster term of conformal PID in the baseline utilizes the relatively accurate Theta model  (which can also be replaced by AR or Transformer) to take advantage of any leftover signal and residualize out systematic errors in the score distribution. It can be regarded as an additional component that “sits on top” of the base forecaster (base model). Therefore, across all test datasets, the performance of conformal PID in the Prophet model consistently outperforms that of the AR and Theta models. This is attributed to the fact that Prophet, being a worse-performing model, is complemented by the superior performance of conformal PID's scorecaster, Theta, which is a better-performing model.
>
>
> >For Table 3: Did the authors try to use a similar scorecaster term within the ECI update and check if that could improve over conformal PID in the Prophet model?
>
> Great point! We maintained the settings of Table 3 to test the performance of ECI combined with the scorecaster (Theta model), and the results (see Table 11) demonstrated that ECI-scorecaster is consistently superior over conformal PID across three base models, thereby validating the effectiveness of the ECI update. Interestingly, for the worse-performing Prophet base model, adding the scorecaster enhanced the performance of ECI, while the scorecaster tended to degrade performance when better-performing base models were used. This observation is also mentioned in Section 2.3 of the conformal PID paper: “an aggressive scorecaster with little or no signal can actually hurt by adding variance to the new score sequence”.
>
>
> >Also, is there any intuition for why decay-OGD baseline beats ECI in Table 5 (Prophet model)?
>
> In the synthetic data of Table 5, there are only three changepoints, and the distribution of data between two changepoints are independent and identically distributed (i.i.d.). After the last changepoint, the distribution of data remains stable and unchanged. Once the optimal value of $q$ is reached, no further adjustments are needed. Hence, the decay mechanism in later stages effectively reduces the learning rate $\eta$, which benefits the performance of decay-OGD.

---

### Author Response · Authors · 2024-11-20

**Main response**

We are grateful to the engaged reviewers, who took a clear interest in the paper and suggested ways to improve it. Thank you!

We hope to respond to the critical comments through the following extensive revisions:

**1.New experiments and detailed interpretations**

We have added new experiments of the Transformer and scorecaster in Appendix F and G.1, respectively. We also revamped Appendix D.1, D.2, and G.2 for detailed discussions on interpretations of the results, and specific hyperparameter tuning ($c$, $w$, and $\eta$, respectively).

**2.Interpretations of the EQ term**

We want to clarify that the EQ term was not intentionally designed but derived based on the idea of smoothed quantile loss.  Further, there are mainly two interpretations for the EQ function’s decreasing when it exceeds a certain level.

Firstly, when $s_t$ is significantly large (namely $Y_t$ is not predicted accurately), according to our method,  the learning rates $\eta_t=\eta \cdot \max_{t-w\leq i,j \leq t}|s_i-s_j|$ also become very large at several subsequent steps. Hence, inaccurate predictions lead to higher learning rates in future steps. In this case, we seek to adjust the interval width by increasing the learning rate, rather than utilizing the EQ term to exaggerate.

Secondly, let’s consider the time series data with the existence of outliers, whose distribution deviates very far from the distribution of normal data. Whenever an outlier appears, it will cause a large error $s_t-q_t$. Note that the distribution of data before and after an outlier remains the same, overemphasizing this outlier can lead to drastic changes in the threshold of interval (for instance, the interval becomes very wide the day after the outlier, but narrows quickly on the third day,  causing excessive oscillations). We believe that overly fluctuating confidence intervals may not be very informative in the context of uncertainty quantification.

**3.Discussion on jags**


Actually, “jags” come from quick reactions to past mistakes. This is the common property of works based on adaptive procedure in conformal inference, and similar phenomenons can be found in ACI, Conformal PID, etc (see Figure 4). If our prediction set correctly covers $Y_t$, we narrow the next prediction interval by a small size (i.e. $\alpha-EQ$). On the contrary, once there is a miscoverage point, we widen the next prediction set by a relatively large value (i.e. $1-\alpha + EQ$) and make the several following points to be covered. Therefore, whenever we widen our interval (due to miscoverage) and then narrow it in the next steps (due to coverages), there is a jag.

Besides, the reason why SF-OGD and decay-OGD seem visually to be more “smooth” is due to the fact that their learning rate is decreasing and close to zero after several iterations, causing the length of predictive intervals to be constant (we can see the jags in the early stages of SF-OGD and decay-OGD in Figure 4). However, when the learning rate is close to zero, the methods cannot react to  sudden distribution shifts.


**4.Proof of variants**

Our proof of both Theorem 1 and Theorem 2 primarily lie in quantifying the EQ term $(s_t-q_t) \nabla f(s_t-q_t)$. Considering ECI-cutoff, all inequalities in the proof still hold after we replace it with the cutoff EQ term $(s_t-q_t) \nabla f(s_t-q_t)$. The assumptions made in Theorem 1 are easier to satisfy. Hence **the proof for ECI-cutoff does not need any adjustment**.

When it turns to the ECI-integral case, ways of measuring inequalities in the original proof must be adjusted according to specific weights. The proof will be more cumbersome. Hence, similar to ACI and Conformal PID, we only prove the ECI form that implies the main mechanism of our method.

**5.Transformer**

The results of the transformers are shown in Appendix F (Tables 9 and 10) of our revision. Consistent with the results of other base models, **ECI and its variants achieve the best performance on five benchmark datasets.**

**6. Scorecaster**

We maintained the same settings to test the performance of ECI combined with the scorecaster (Theta model), and the results (see Table11) demonstrated that **ECI+scorecaster is consistently superior over Conformal PID across three base models**, thereby validating the effectiveness of the ECI update. Interestingly, for the worse-performing base model (Prophet), adding the scorecaster enhanced the performance of ECI, while the scorecaster tended to degrade performance when better-performing base models were used. This observation is also mentioned in Section 2.3 of the Conformal PID paper: “an aggressive scorecaster with little or no signal can actually hurt by adding variance to the new score sequence”.

---

### Meta-Review · Area_Chair_rRGe · 2024-12-22

**Metareview:**

This paper proposes Error-quantified Conformal Inference (ECI), an extension of online conformal inference that incorporates the magnitude of miscoverage errors into its update procedure through a smoothed quantile loss function. By leveraging this additional feedback, ECI adapts more effectively to distributional shifts in time series data, producing narrower prediction intervals while maintaining coverage guarantees. The authors provide theoretical results supporting the validity of ECI and its variants, as well as empirical evaluation on synthetic and real-world datasets.

**Additional Comments On Reviewer Discussion:**

The reviewers generally recognized the novelty and effectiveness of the proposed method, particularly its handling of dynamic miscoverage errors in the context of time series. While some concerns were raised regarding hyperparameter tuning, the jagged nature of intervals, and the novelty of the work, the authors addressed these points in their rebuttal. Although a few reviewers highlighted derivative aspects of the work, the consensus leaned towards acceptance due to its methodological contributions and empirical results.

---

### Decision · Program_Chairs · 2025-01-22

Accept (Poster)